# Design and validation of a foldable and photovoltaic wide-field epiretinal prosthesis

Laura Ferlauto[1], Marta Jole Ildelfonsa Airaghi Leccardi [1], Naïg Aurelia Ludmilla Chenais[1], Samuel Charles Antoine Gilliéron[1], Paola Vagni[1], Michele Bevilacqua[1], Thomas J. Wolfensberger[2], Kevin Sivula [3] & Diego Ghezzi [1]

Retinal prostheses have been developed to fight blindness in people affected by outer retinal layer dystrophies. To date, few hundred patients have received a retinal implant. Inspired by intraocular lenses, we have designed a foldable and photovoltaic wide-field epiretinal prosthesis (named POLYRETINA) capable of stimulating wireless retinal ganglion cells. Here we show that within a visual angle of 46.3 degrees, POLYRETINA embeds 2215 stimulating pixels, of which 967 are in the central area of 5 mm, it is foldable to allow implantation through a small scleral incision, and it has a hemispherical shape to match the curvature of the eye. We demonstrate that it is not cytotoxic and respects optical and thermal safety standards; accelerated ageing shows a lifetime of at least 2 years. POLYRETINA represents significant progress towards the improvement of both visual acuity and visual field with the same device, a current challenging issue in the field.

[1] Medtronic Chair in Neuroengineering, Center for Neuroprosthetics, Institute of Bioengineering, School of Engineering, École Polytechnique Fédérale de Lausanne, Lausanne, Switzerland. [2] Department of Ophthalmology, University of Lausanne, Hôpital Ophtalmique Jules-Gonin, Fondation Asile des Aveugles, Lausanne, Switzerland. [3] Laboratory for Molecular Engineering of Optoelectronic Nanomaterials, Institute of Chemical Sciences and Engineering, School of Basic Science, École Polytechnique Fédérale de Lausanne, Lausanne, Switzerland. Laura Ferlauto, Marta J. I. Airaghi Leccardi and Naïg A. L. Chenais contributed equally to this work. Correspondence and requests for materials should be addressed to D.G. (email: diego.ghezzi@epfl.ch)

Blindness affects more than 30 million people worldwide[1], and it is defined as visual acuity of less than 20/400 or a corresponding visual field loss to less than 10 degrees, in the better eye with the best possible correction[2]. In North America and most of European countries, legal blindness is defined as visual acuity of 20/200 or visual field no greater that 20 degrees. In the last decade, various visual prostheses have been developed to fight blindness in case of retinal dystrophies, such as Retinitis pigmentosa[3] and more recently age-related macular degeneration (Clinical Trial NCT02227498). Several multi-center clinical trials showed the feasibility of restoring a coarse form of vision with retinal implants, such as single letters discrimination and simple objects recognition[4,5]. However, several challenges remain open, such as the improvement of visual acuity and the enlargement of the visual field above the thresholds of blindness[6]. An agreed upon strategy to improve visual acuity is to increase the electrode density, while a large visual field could be attained by enlarging the retinal coverage with a larger prosthesis.

Concerning the visual field, tests on healthy subjects under pixelated vision indicated that an array of $25 \times 25$ pixels and 30 degrees of visual angle (about 8.5 mm in diameter) could provide adequate mobility skills[7,8]. However, the size of the prosthesis is typically limited by the maximal allowed sclerotomy, which is about 6–7 mm long; available prostheses are therefore in the range of 1 to 5 mm. Argus II™, the largest implanted electrode array in humans, is a $6 \times 10$ array with a 575 μm electrode pitch[4] and a theoretical field of view of $10 \times 18$ degrees. Increasing the size of the array is associated with two main challenges: it requires a large scleral incision and it may not conform to the eye curvature. In a flat prosthesis placed over the retina, central and peripheral electrodes may not be at the same distance from the retina. A large distance will inevitably increase the stimulation threshold and the cross-talk between adjacent electrodes[9]. Preliminary attempts in designing wide-field retinal prosthesis have been proposed[9,10]. However, these approaches are based on materials (i.e., polyimide) with high elastic modulus (GPa), very thin substrates (e.g., 10 μm), and complex shapes (e.g., star) that could create challenges in manipulation, implantation, and fixation.

Concerning visual acuity, previous researches estimated that, to be useful in daily life, a retinal prosthesis should have 500 pixels distributed in the central area of approximately 5 mm in diameter[11,12]. More recently, a trial on healthy subjects showed that the number of pixels required to recognize common objects is on the order of 3000–5000[13]. Despite microfabrication techniques allow such electrode density, a limitation remains due to the routing of the connection tracks in the active area and the size of the flat cable connection to the implantable electronics/stimulator. To overcome these issues, in photovoltaic stimulation[14], the light projected into the pupil is wirelessly converted into electrical stimuli delivered to the retina. After the first demonstration of vision restoration in blind rats with a silicon photovoltaic subretinal prosthesis[15], a second major step was achieved with the exploitation of conjugated polymers and organic semiconductors (i.e., poly(3,4-ethylenedioxythiophene)-poly(styrenesulfonate), PEDOT:PSS; regioregular poly(3-hexylthiophene-2,5-diyl), P3HT; [6,6]-phenyl-C61-butyric acid methyl ester, PCBM) to build an organic photovoltaic subretinal interface[16–18]. In the latter, despite the capability of improving visual acuity in dystrophic rats after 1 month of implantation[19], several issues remain unsolved. Conjugated polymers are well tolerated when exposed to the subretinal space[18], but they start to delaminate a few months after placement leading to an unavoidable degradation of the organic materials. Moreover, in the cases of both silicon and organic photovoltaic subretinal prostheses, the limited size of the devices (1–2 mm) will not allow the recovery of a large visual field, unless implanting multiple devices[20]. Some concerns remain about the risks associated with the implantation of multiple devices in the subretinal space (e.g., retinal detachment, movements of the devices, and device overlaps). Thus, increasing both visual acuity and visual field size with a single retinal prosthesis remains one of the main unsolved challenges in the field[11].

## Results

**Design and fabrication.** POLYRETINA is a novel foldable and photovoltaic wide-field epiretinal prosthesis based on poly (dimethylsiloxane) (PDMS) as substrate material, because of its transparency, elasticity, low Young's modulus, and high strain to failure[21,22]. Moreover, PDMS is available as medical grade elastomer already in use in medical device applications. The device consists in a PDMS–photovoltaic interface (Fig. 1a, c), embedding 2215 stimulating pixels (80 and 130 μm in diameter) distributed on an active area of 12.7 mm (Supplementary Fig. 1a). Each pixel is composed by a PEDOT:PSS bottom anode, a P3HT:PCBM (referred also as Blend) semiconductor layer, and a top cathode in titanium (Ti). Another PDMS layer encapsulates the prosthesis, avoiding the delamination and degradation of the organic materials and extending its lifetime (Supplementary Fig. 1b). Openings of 67 and 120 μm in diameter have been made in the encapsulation layer to expose the cathodes (Fig. 1f). Titanium is a mechanically and electrochemically stable metal, it is widely used in implantable devices, it has an appropriate work function for the photovoltaic mechanism, and it is a capacitive charge-injection material (also due to the thin layer of titanium oxide formed at the surface). The latter is desirable with mono-phasic pulses, as in this photovoltaic approach, because no chemical species are created or consumed during a stimulation pulse[23], thus avoiding undesired tissue reactions. Under this condition, the electrode/electrolyte interface can be modeled as pure electrical capacitor without electron transfer from the metal to the solution[24]. To verify this hypothesis, we measured the pH with a microelectrode positioned above the titanium electrode of the PDMS–photovoltaic interface (Supplementary Fig. 2) upon 1 h of pulsed illumination (20 Hz, 10 ms, 3.4 mW mm$^{-2}$; $N = 3$ devices). The irradiance has been set to a value above the maximum allowed for prosthetic application (see Optical and thermal safety). During illumination, a negligible pH shift of about 0.002 pH units has been detected, which could be explained by a recording artifact due to the local temperature increase induced by the prosthesis (see Optical and thermal safety). Local heating could reduce the resistivity of the solution and decrease the voltage difference between the pH microelectrode and the local reference.

The hemispherical shape of POLYRETINA (Fig. 1b, d) is obtained by bonding the PDMS–photovoltaic interface on a dome-shaped PDMS support (Fig. 1a) with a radius of curvature of 12 mm, corresponding to the standard human eye. The bonding induces a radial elongation in the PDMS–photovoltaic interface of about 3% (in diameter), which has been considered to determine the covered retinal surface (Supplementary Fig. 1c). Four anchoring wings, with holes for tacks, have been included for the fixation of the prosthesis (Fig. 1d). The folding of POLYRETINA, its insertion, and covering of the retinal surface have been evaluated in simulated surgeries with plastic models of the human eye (Fig. 2a). The prosthesis can be folded prior implantation (Figs. 1e, 2a, top-left), inserted through an aperture of 6.5 mm (Fig. 2a, top-right), released within the posterior chamber (Figs. 1g, 2b, bottom-right and bottom-left), and attached in epiretinal configuration (Fig. 2b). The same surgical approach has been also validated in enucleated pig eyes (Fig. 2c).

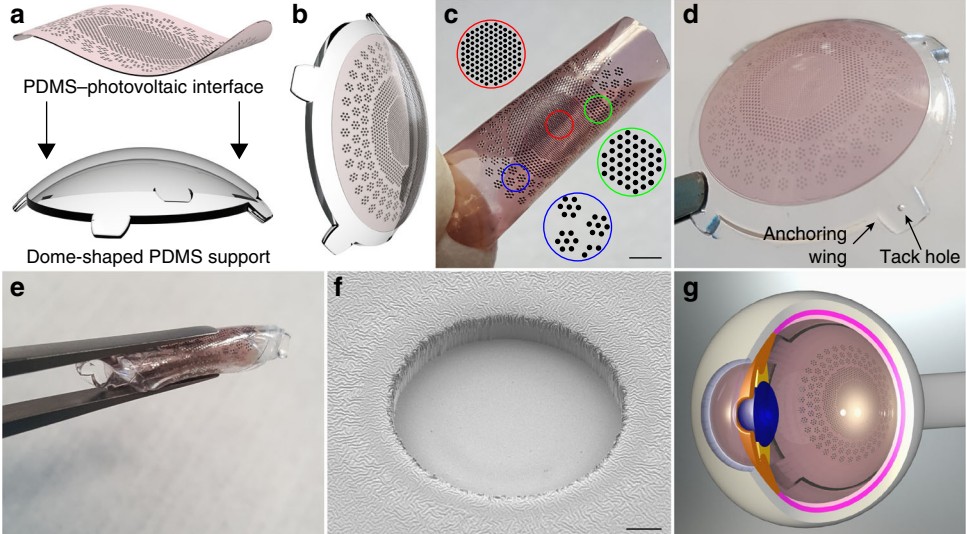

**Fig. 1** Foldable and photovoltaic wide-field retinal prosthesis. **a** 3D model of the fabricated PDMS-interface and of the dome-shaped PDMS support. **b** 3D model of the retinal prosthesis after boding the PDMS-interface to the PDMS support. **c** Fabricated PDMS–photovoltaic interface with pixels arranged in three areas of different sizes and densities: central area (red), diameter of 5 mm, 967 electrodes in hexagonal arrangement, electrode diameter 80 μm and pitch 150 μm, density 49.25 px mm$^{-2}$; first ring (green), diameter of 8 mm, 559 electrodes in hexagonal arrangement, electrode diameter 130 μm and pitch 250 μm, density 17.43 px mm$^{-2}$; second ring (blue), diameter 12.7 mm, 719 electrodes, electrode diameter 130 μm, density 9.34 px mm$^{-2}$. Circles show an enlarged view of the pixels distribution. Scale bar is 2.5 mm. **d** Picture of POLYRETINA. Four anchoring wings with holes are present for attaching the prosthesis with retinal tacks. **e** POLYRETINA folded before injection. **f** Scanning electron microscope image (40° tilted view) of a photovoltaic pixel. Scale bar is 10 μm. **g** 3D model after epiretinal placement

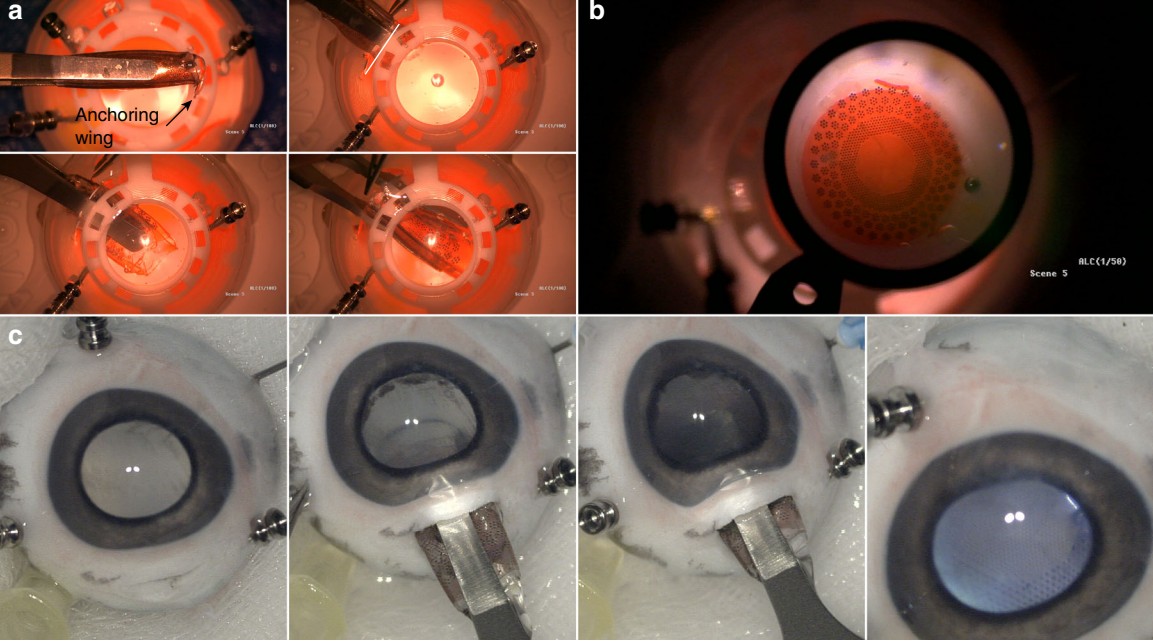

**Fig. 2** Simulated surgical implantation. **a** Picture sequence of the implantation in a human eye plastic model. The white line in top-right panel shows the incision of 6.5 mm. **b** Picture of POLYRETINA placed in epiretinal configuration. **c** Picture sequence of the implantation in a pig eye

**Optimization of the photovoltaic pixel**. First, using Kelvin probe force microscopy (KPFM), we evaluated the changes in the surface potential generated at the cathode upon illumination for different conditions of fabrication (Fig. 3a, b). To assess the role of the bottom anode, we fabricated photovoltaic interfaces onto glass substrates including a bottom anode made of indium tin oxide (ITO), an injection layer of PEDOT:PSS, a semiconductor layer of P3HT:PCBM, and aluminum (Al) top cathodes. We initially used aluminum since it is one of the most common cathode material in organic photovoltaics. KPFM measures (Fig. 3c) across several devices showed that the variation of the surface potential upon illumination (white LED, light from the top, 0.4 mW mm$^{-2}$) is about 15 folds higher (Fig. 3d) with aluminum cathodes with respect to P3HT:PCBM only. When

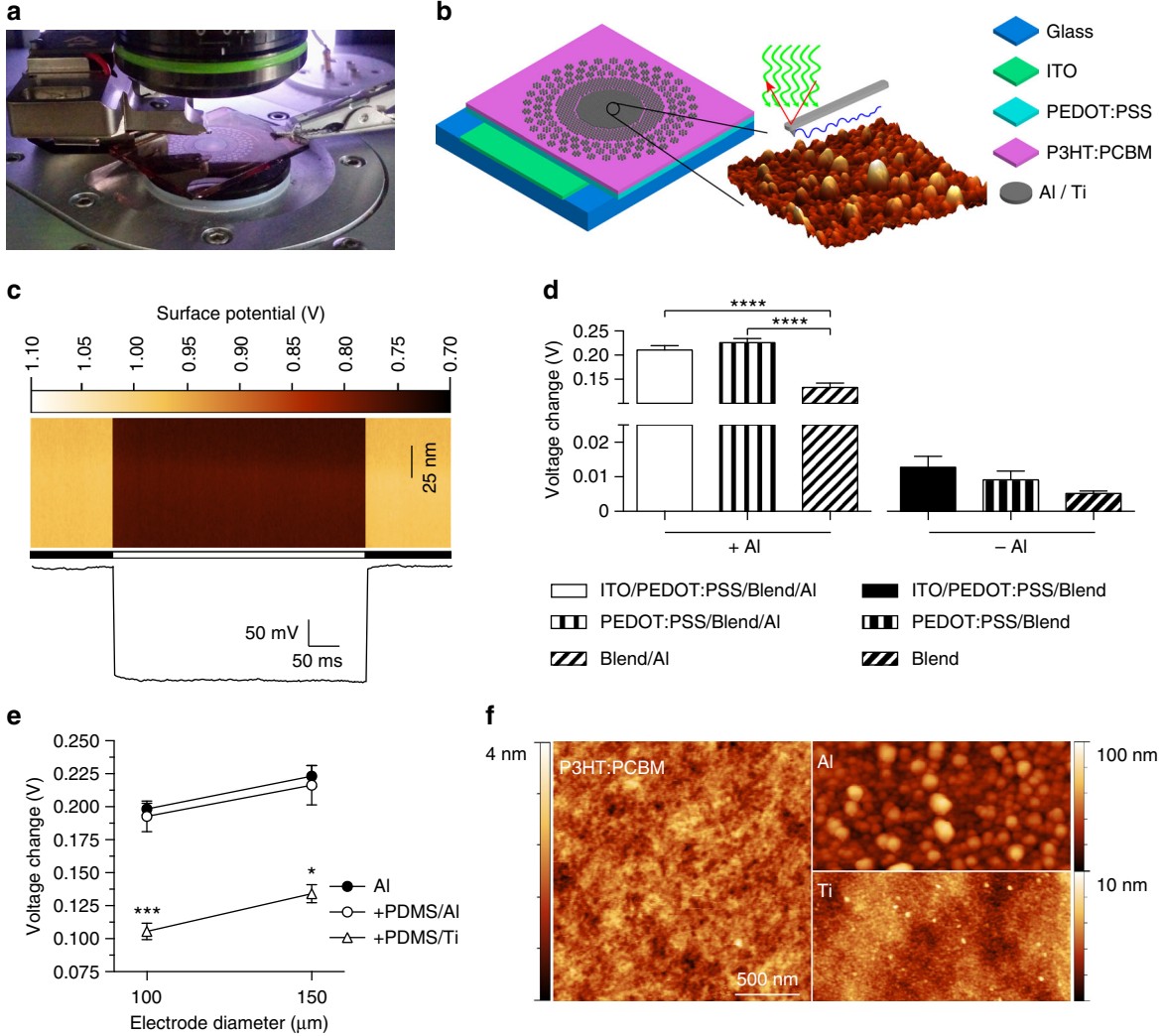

**Fig. 3** Optimization of the photovoltaic pixel. **a** Picture of the KPFM measures. **b** Sketch of the fabricated device. Glass substrates have been coated with a thin film of ITO (200 nm), a thin film of PEDOT:PSS (50 nm), a thin film of P3HT:PCBM (100 nm), and last aluminium (100 nm) or titanium (150 nm). **c** Representative KPFM map on a Glass/PEDOT:PSS/Blend/Al device obtained by repeating a line scan of 100 nm (vertical direction). The horizontal bar indicates period of dark (black) and light (white). The bottom panel shows the average potential fluctuation during time; each point is the average potential in a single line scan. **d** Surface potential variations (voltage in light—voltage in dark) for 6 different architectures. Each bar is the mean (±s.e.m.) of at least $N = 3$ devices, in which at least $n = 3$ electrodes/points has been measured and averaged. ITO/PEDOT:PSS/Blend/Al: 0.2106 ± 0.0092 V, $N = 5$, $n = 3$; PEDOT:PSS/Blend/Al: 0.2259 ± 0.0085 V, $N = 5$, $n = 3$; Blend/Al: 0.1334 ± 0.0090 V, $N = 3$, $n = 3$; ITO/PEDOT:PSS/Blend: 0.0128 ± 0.0032 V, $N = 3$, $n = 3$; PEDOT:PSS/Blend: 0.0091 ± 0.0025 V, $N = 3$, $n = 4$; Blend: 0.0052 ± 0.0007 V, $N = 3$, $n = 4$. One-way ANOVA, $p < 0.0001$, $F = 177.9$. **e** Surface potential variations with/without a bottom PDMS layer and with Al or Ti top contacts of 100 and 150 μm in diameter. Each point is the mean (±s.e.m.) of at least $N = 3$ devices, in which at least $n = 3$ electrodes has been measured and averaged. PEDOT:PSS/Blend/Al-100 μm: 0.1984 ± 0.0043 V, $N = 3$, $n = 3$; PEDOT:PSS/Blend/Al-150 μm: 0.2232 ± 0.0082 V, $N = 3$, $n = 3$; PDMS/PEDOT:PSS/Blend/Al-100 μm: 0.1927 ± 0.0115 V, $N = 5$, $n = 3$; PDMS/PEDOT:PSS/Blend/Al-150 μm: 0.2163 ± 0.0150 V, $N = 5$, $n = 3$; PDMS/PEDOT:PSS/Blend/Ti-100 μm: 0.1055 ± 0.0063 V, $N = 3$, $n = 6$; PDMS/PEDOT:PSS/Blend/Ti-150 μm: 0.1342 ± 0.0068 V, $N = 3$, $n = 3$. **f** Representative AFM images of PEDOT:PSS/Blend, PEDOT:PSS/Blend/Al, and PEDOT:PSS/Blend/Ti surfaces

aluminum is present (Fig. 3d, left), the absence of any anode (ITO or ITO/PEDOT:PSS) significantly reduces the surface potential variation upon illumination (ITO/PEDOT:PSS/Blend/Al vs. Blend/Al, $p < 0.0001$; PEDOT:PSS/Blend/Al vs. Blend/Al, $p < 0.0001$; one-way ANOVA, Tukey's multiple comparison test). No significant difference has been found with or without the ITO anode if the PEDOT:PSS injection layer is present (ITO/PEDOT:PSS/Blend/Al vs. PEDOT:PSS/Blend/Al, $p = 0.6219$; one-way ANOVA, Tukey's multiple comparison test). In the absence of aluminum cathodes (Fig. 3d, right), the architectures with different bottom anodes do not induce any significant difference (ITO/PEDOT:PSS/Blend vs. PEDOT:PSS/Blend, $p = 0.9997$; ITO/

PEDOT:PSS/Blend vs. Blend, $p = 0.9890$; PEDOT:PSS/Blend vs. Blend, $p = 0.9995$; one-way ANOVA, Tukey's multiple comparison test). The maximization of the surface potential variation has been obtained with aluminum cathodes and both ITO/PEDOT:PSS or only PEDOT:PSS anodes. Therefore, to simplify the fabrication process, we implemented PEDOT:PSS alone as bottom layer. We also verified that the surface potential variation was not altered (Fig. 3e) when the interface was built over a PDMS substrate instead of bare glass with aluminium cathode diameters of both 100 and 150 μm (● PEDOT:PSS/Blend/Al and ○ PDMS/PEDOT:PSS/Blend/Al); no statistical differences have been found among the groups (two-way ANOVA, Tukey's multiple

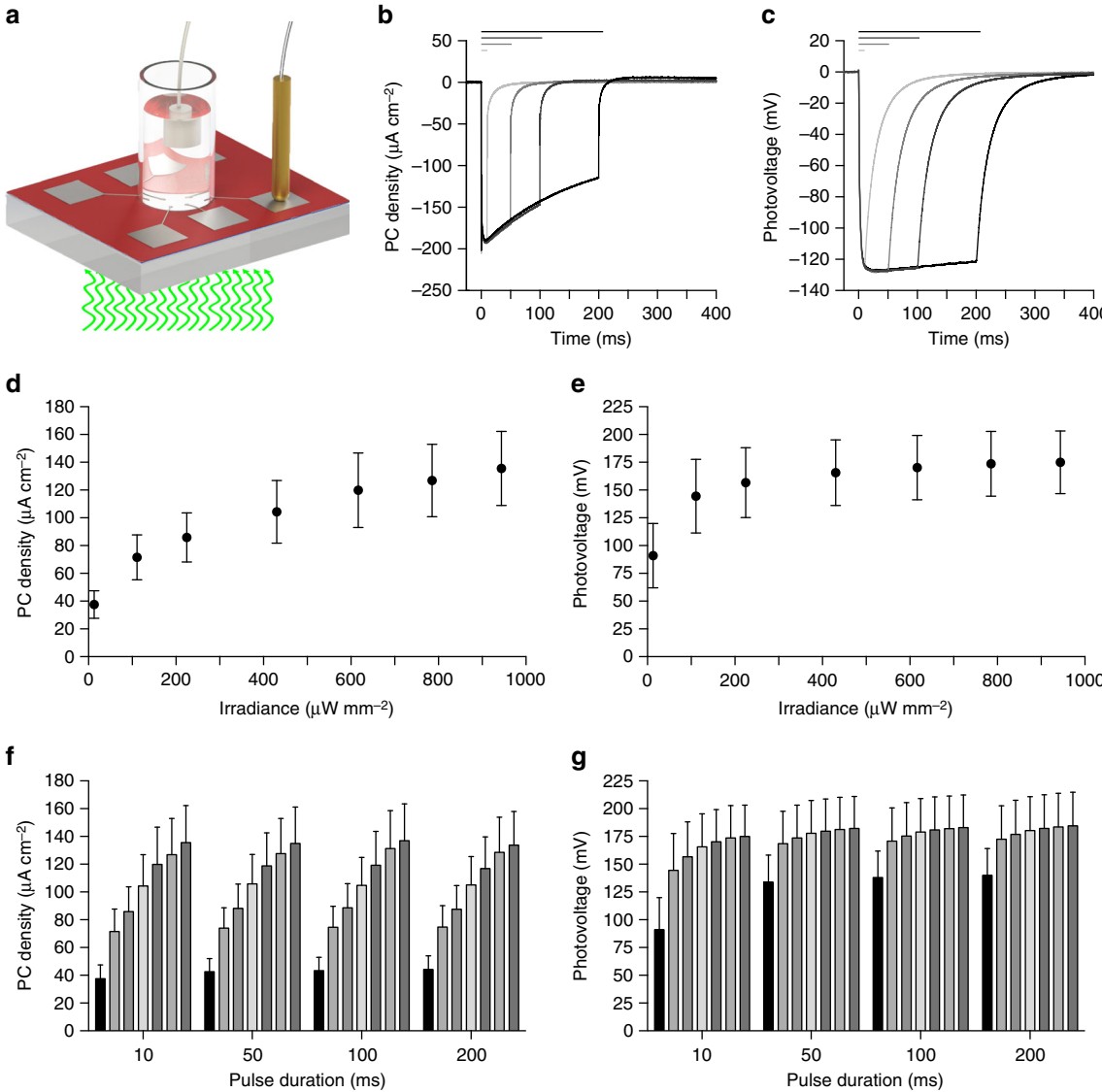

**Fig. 4** Characterization of the photo-current and photo-voltage. **a** Drawing of the experimental setup for the measure of PC and PV; the light pulse comes from the bottom. **b**, **c** Examples of PC density (**b**) and PV (**c**) measures obtained from 1 electrode (diameter 100 μm) at maximal light intensity (565 nm, 943.98 μW mm$^{-2}$) and for increasing pulse durations (10, 50, 100, and 200 ms). Horizontal bars represent the light pulses. **d,e**, Mean (±s.e.m) PC density (**d**) and PV(**e**) measured upon illumination with 10 ms pulses at increasing light intensities. **f**, **g**, Mean (±s.e.m) PC density (**f**) and PV (**g**) measured for increasing light intensities (12.75, 111.11, 225.00, 430.56, 616.67, 785.65, and 943.98 μW mm$^{-2}$) and pulse durations (10, 50, 100, and 200 ms). In panels **d** to **g**, the PC density and PV on every device ($N = 3$) has been measured for all electrodes ($n = 6$) and data have been averaged

comparison test, interaction $p = 0.9633$; factor 1, diameter, $p = 0.0887$; factor 2, substrate, $p = 0.6385$). When titanium replaces aluminium (Δ PDMS/PEDOT:PSS/Blend/Ti), the surface potential is slightly reduced (for 100 μm: one-way ANOVA, $F = 25.43$, $p < 0.001$; PDMS/PEDOT:PSS/Blend/Ti vs. both PEDOT:PSS/Blend/Al and PDMS/PEDOT:PSS/Blend/Al, $p < 0.001$, Tukey's multiple comparison test; for 150 μm: one-way ANOVA, $F = 9.266$, $p < 0.01$; PDMS/PEDOT:PSS/Blend/Ti vs. both PEDOT:PSS/Blend/Al and PDMS/PEDOT:PSS/Blend/Al, $p < 0.05$, Tukey's multiple comparison test).

KPFM measurements have been performed in air in non-contact mode; therefore, the measured variations in the surface potential may be slightly different with respect to the electric potential generated by the double layer capacitive charging occurring at an electrode–electrolyte interface, as in the case of an implanted retinal prosthesis. Therefore, we measured the photo-

current (PC) and the photo-voltage (PV) generated in the presence of electrolyte solution upon illumination. We fabricated chips embedding six electrodes, each of them connected to a contact pad for measuring the signal with respect to an Ag/AgCl reference electrode immersed in solution (Fig. 4a). Both PC and PV have been measured with illumination (565 nm) at increasing light intensities (12.75, 111.11, 225.00, 430.56, 616.67, 785.65, and 943.98 μW mm$^{-2}$) and pulse duration (10, 50, 100, and 200 ms). The PC (Fig. 4b) generated by pulsed illumination (943.98 μW mm$^{-2}$) has a typical capacitive profile, peaking in about 10 ms and then decreasing with an exponential decay, while the PV (Fig. 4c) reaches a steady-state value and remains constant. This is in agreement with the capacitive nature of the electrode/ electrolyte interface. Moreover, the PV generated (about 180 mV) is largely below the redox potential of titanium (or titanium oxide), thus ensuring that no irreversible reactions occur at the

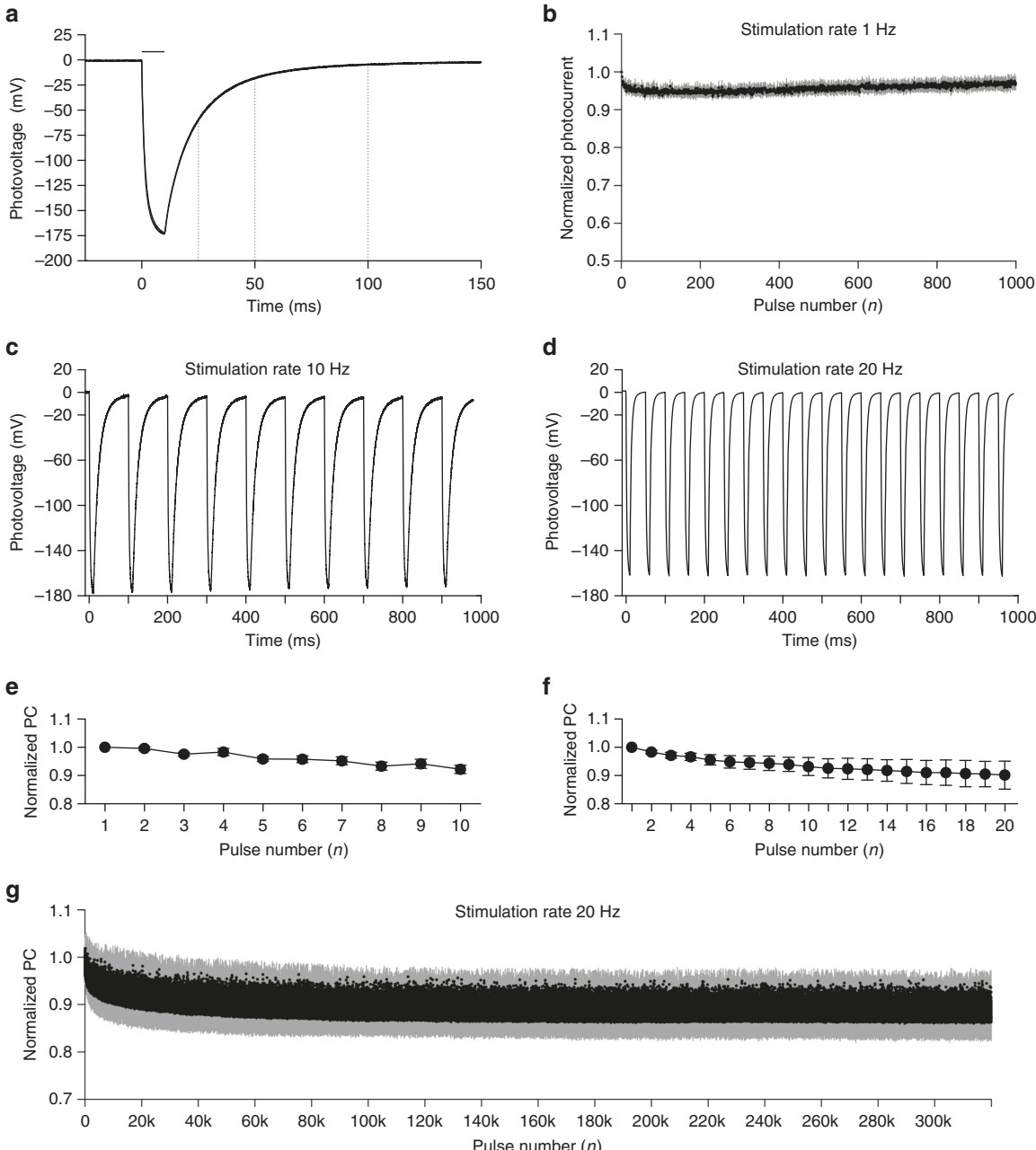

**Fig. 5** High-frequency train stimulation. **a** Mean PV trace obtained at maximal light intensity (565 nm, 10 ms, 943.98 µW mm⁻²). The trace is the mean of $N = 6$ devices; in which $n = 6$ electrodes have been measured and averaged. The horizontal bars represent the light pulse. The dotted lines highlight the discharging rate of the electrode. **b** Evolution of the PC density peaks during 1000 stimuli delivered at 1 Hz (10 ms, 943.98 µW mm⁻²). Each point is the mean (±s.e.m.) of $N = 3$ devices, in which $n = 6$ electrodes have been measured and averaged. **c** Representative PV recording upon 10 pulses at 10 Hz (565 nm, 10 ms, 943.98 µW/mm²). **d** Representative PV recording upon 20 pulses at 20 Hz (565 nm, 10 ms, 943.98 µW mm⁻²). **e** Evolution of the PC density peaks normalized to the first pulse. Each point is the mean ± s.e.m. of $N = 10$ devices, in which $n = 6$ electrodes have been measured and averaged. **f** Evolution of the PC density peaks normalized to the first pulse. Each point is the mean ± s.e.m. of $N = 8$ devices, in which $n = 6$ electrodes have been measured and averaged. **g** PC generated with 320,000 stimuli delivered at 20 Hz (565 nm, 10 ms, 943.98 µW mm⁻²). Each point is the mean ± s.d. of $n = 2$ electrodes from $N = 1$ device

interface. The PC density increases with irradiance, with a mean (±s.e.m.) peak value of $135.51 \pm 26.74\ \mu\text{A cm}^{-2}$ (10 ms) for 943.98 µW mm⁻² (Fig. 4d, f). According to the literature in the field[22], these current values should be able to induce epiretinal stimulation of retinal ganglion cells (RGCs). The slope of the PC density profile is decreasing while increasing irradiance, and a saturation of the response could be expected for irradiance higher than 1–2 mW mm⁻². We also measured the PC density (10 ms, 943.98 µW mm⁻²) after 48 h of immersion in physiological

solution (stored in dark). The mean (±s.e.m.) ratio before/after has been measured in $94.44 \pm 12.28$, $95.11 \pm 13.07$, $93.36 \pm 13.26$, $94.99 \pm 12.48\%$, respectively, for 10, 50, 100, and 200-ms pulses; no significant differences have been found (10 ms: $p = 0.4423$; 50 ms: $p = 0.5798$; 100 ms: $p = 0.5798$; 200 ms: $p = 0.5526$; $N = 3$ devices, $n = 6$ electrodes per device; Wilcoxon matched-pairs signed rank test).

Ti-based photovoltaic electrodes show a full discharge (97.7%) after 100 ms (Fig. 5a) when illuminated with 10-ms pulses

(943.98 μW mm$^{-2}$); while they are discharged of 65.4 and of 89.9% after 25 ms and 50 ms respectively. This suggests that POLYRETINA could operate in the 1–20 Hz range without the need of an external shunting resistor[25]. To characterize the stimulation efficiency over repetitive stimuli, we measured the PC over 1000 stimuli (Fig. 5b) delivered at 1 Hz (10 ms, 943.98 μW mm$^{-2}$). The mean (±s.e.m.) steady state response (average of the last 20 pulses/first response) is almost unchanged (96.99 ± 1.51%). At a higher stimulation frequency, such as 10 Hz, the electrodes are entirely discharged between pulses (Fig. 5c), therefore the PC density is not largely affected by repetitive stimulations; in a train of 10 pulses at 10 Hz, the mean (±s.e.m.) ratio last/first responses is 92.20 ± 1.52 % (Fig. 5e). Also, in a train of 20 pulses at 20 Hz, the mean (±s.e.m.) ratio last/first responses is 90.21 ± 4.96 % (Fig. 5f). Given the possibility to stimulate at 20 Hz, we tested Ti-based photovoltaic electrodes over a long operation period (Fig. 5g). Upon 320,000 stimuli (20 Hz, 10 ms, 943.98 μW mm$^{-2}$), the stable steady state response (average of the last 1000 pulses/first response) is only slightly affected (88.6%).

**Validation ex vivo with explanted retinas from blind mice.** Next, we tested the ex vivo efficacy of the PDMS–photovoltaic interface in stimulating RGCs. For this purpose, we used the retinal degeneration 10 mouse model[26], that is recognized as an excellent model for Retinitis pigmentosa[27]. Extracellular recordings of prosthetic-evoked spiking activity of RGCs have been collected from retinas explanted from old mice to avoid as much as possible the natural responses from surviving photoreceptors ($n = 39$ cells, $N = 15$ mice; mean ± s.d. age 140.87 ± 20.35 days). Retinas have been layered on the central 5-mm area of the PDMS–photovoltaic interface mimicking the epiretinal configuration (Fig. 6a). According to the PC density measures, we tested only 10-ms pulses (peak of the PC response) with a broad range of irradiance (from 47.35 to 29.07 mW mm$^{-2}$). Light pulses induced a prosthetic-evoked spiking activity in the recorded RGC (Supplementary Fig. 3a and Fig. 6b). Spikes have been detected with a threshold algorithm (red lines in Fig. 6b and Supplementary Fig. 3a), converted into a raster plot (Supplementary Fig. 3a, middle), and presented as peri-stimulus time histogram (PSTH; Supplementary Fig. 3a, bottom). As previously reported[28], we observed three types of responses, classified as short, medium, and long latency (SL, ML, and LL). The presence of SL spikes (elicited in the 10-ms window after the light onset, 1 bin) indicates a direct electrical stimulation of RGCs; while the presence of ML and LL spikes indicates a network-mediated activation. We have found that SL spikes can be evoked starting from the first irradiance tested (47.35 μW mm$^{-2}$), then the firing rate slowly increases and it remains stable above 1.08 mW mm$^{-2}$ till the highest irradiance tested (Fig. 6c and Supplementary Fig. 3c). However, the mean (± s.e.m.) latency (Fig. 6d) at this first irradiance is relatively long (6.05 ± 0.23 ms); it decreases with the increase of the irradiance, and it stabilizes at 4.12 ± 0.07 ms for irradiance higher than 1.08 mW mm$^{-2}$ (Fig. 6d and Supplementary Fig. 3d). In this range (higher than 1.08), the mean (± s.e.m.) jitter of the first SL spike is 0.39 ± 0.05 ms. This suggests that the SL response is saturated for irradiance higher than 1.08 mW mm$^{-2}$, as predicted by the measure of the PC densities. For irradiance lower than 1.08 mW/mm$^2$ the mean latency appears shorter but the jitter is more variable, indicating a more instable response (Fig. 6d). The firing rate of ML (Fig. 6e and Supplementary Fig. 3e) and LL (Fig. 6f and Supplementary Fig. 3f) spikes growth more progressively, but they also become stable after 1.08 mW mm$^{-2}$. As a control, when retinas have been layered on bare PDMS substrates ($n = 34$, $N = 13$; 143.08 ± 32.09 days), no light-evoked responses have been detected for all

the irradiance tested (Supplementary Fig. 3b and Supplementary Fig. 4b). As already demonstrated by others[29], we also verified in a second subset of cells ($n = 6$, $N = 5$; 209.4 ± 37.14 days) that the prosthetic activation of both ML and LL spikes is abolished by using synaptic blockers (Supplementary Fig. 5). This confirms the hypothesis that ML and LL are induced by the activation of the internal retinal circuit.

**Spatial selectivity.** We then addressed the spatial selectivity by using an experimental/computation hybrid approach. First, using a glass microelectrode (Fig. 7a, b) we measured the radial voltage spreading in three directions (D1, D2, and D3) upon illumination of a single pixel (Fig. 7c). For each illuminated pixel ($n = 4$ pixels), the normalized voltage spreading in the three principal directions have been averaged. The mean (±s.e.m.) voltage distribution across all the pixel tested has been plotted and interpolated with a Gaussian function (Fig. 7d). Experimental data match with the normalized voltage profile obtained by a finite element analysis (FEA) model (Fig. 7d, dotted blue line). The full width at half maximum (FWHM) of the simulated curve (Fig. 7d, dotted gray line) has been taken as the effective activation area, which is about 100 μm. FEA simulations have been used to characterize the normalized voltage profile induced by illumination of increasing diameters (Fig. 7e). Increasing the spot size from 1 pixel to 7 and 19 pixels increases the potential. Last, we simulate the effect of different patterns of activation (Fig. 7f). In all cases, a spatially selective potential profile corresponding to light pattern is shown.

**Cytotoxicity and long-term functioning.** To validate the long-term functioning of POLYRETINA, we tested the mechanical impact of the hemispherical shape. For this purpose, the PDMS–photovoltaic interface has been bonded on the dome-shaped PDMS support. The bonding procedure induces tensile stresses in the PDMS–photovoltaic interface leading to the formation of cracks in the polymers and the titanium cathodes (Fig. 8a, top row). To avoid cracks in the titanium cathodes, SU-8 rigid platforms[30] have been integrated below each cathode in the substrate of the interface (Supplementary Fig. 1b). With this precaution, the pixel above the SU-8 rigid platforms is protected from cracks (Fig. 8a, bottom row); images correspond to the green area in Fig. 1c. Cracks are still formed within the blend film in the area between SU-8 rigid platforms, however this is less critical since that area is encapsulated in PDMS to prevent delamination and the carriers photo-generated outside of the area defined by the cathode do not significantly contribute to the photo-potential/current generated at the electrode/electrolyte interface. Then, we measured the changes in the surface potential by using KPFM (Fig. 8b). Due to the hemispherical shape, only the electrodes at the top of the prosthesis (80 μm in diameter / 67 μm openings) can be reached by the AFM tip. The surface potential change induced by illumination (white LED, light from the top, 0.4 mW mm$^{-2}$) is not statistically different (Mann–Whitney test, $p = 0.8182$) with respect to the planar PDMS-interface (Fig. 8c).

To simulate the lifetime of POLYRETINA once implanted, we performed a functional accelerating ageing test by immersion in physiological saline solution hold at 87 °C (Fig. 8d). The changes of the surface potential upon illumination have been measured with KPFM before starting the ageing and at several time points during the protocol (Fig. 8e). No statistically significant changes in the mean (±s.d.) surface potential have been detected till 24 months of accelerated ageing (one-way ANOVA, $F = 0.1252$, $p = 0.9731$). Last, according to ISO 10993-5: Biological Evaluation of Medical Devices, in vitro cytotoxicity has been evaluated via an

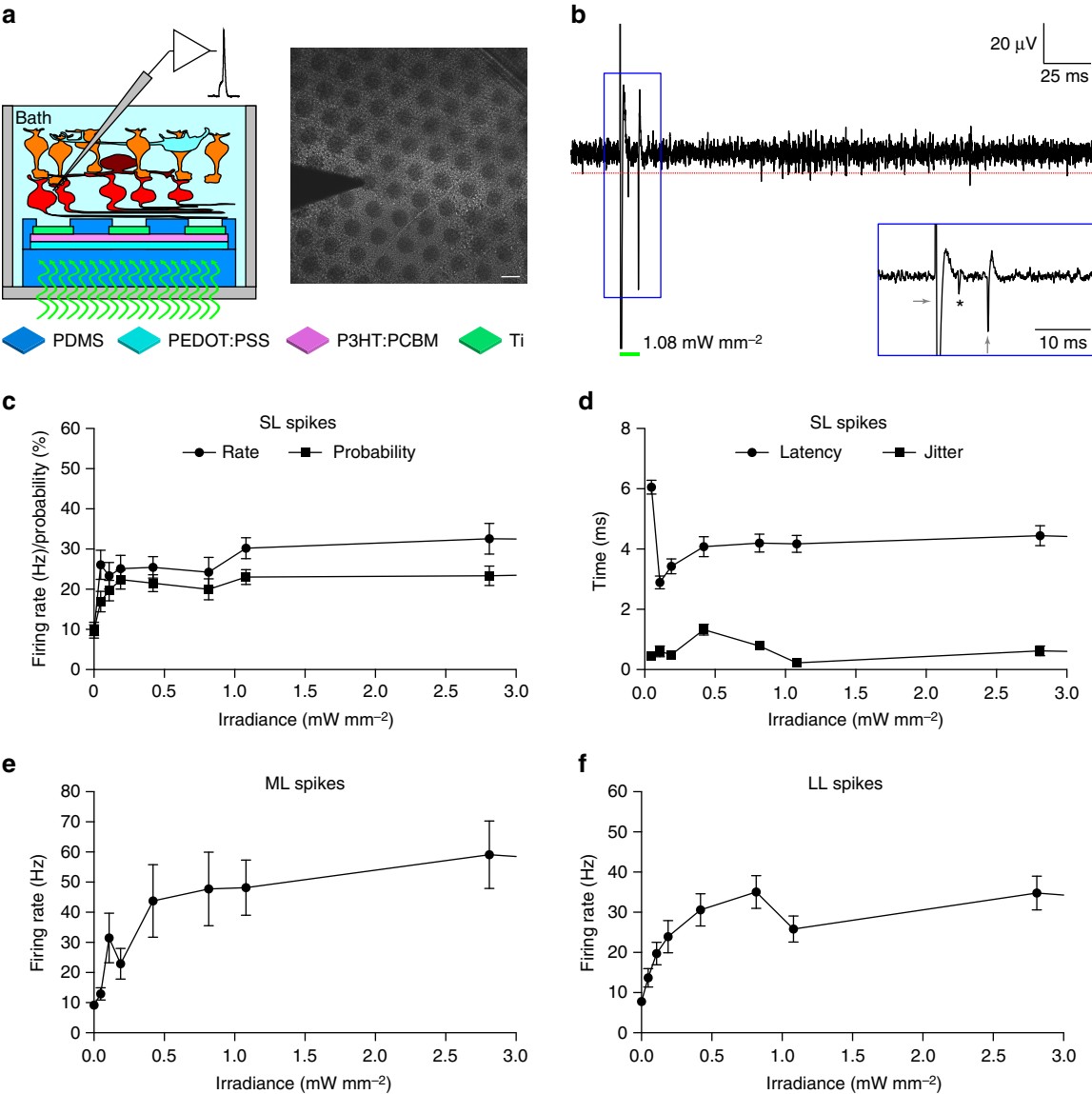

**Fig. 6** Evaluation ex vivo with retinal explants. **a** Sketch of the recording set-up together with a picture of a retinal explant over the PDMS–photovoltaic interface with the metal electrode used for recordings. Scale bar is 100 μm. **b** Representative single-sweep recording from a retinal ganglion cell over PDMS–photovoltaic interface upon 10-ms illumination at 1081.7 μW mm$^{-2}$. The red dotted line is the threshold set for spike detection. The green bar represents the light pulse. The blue insert shows a magnification of the period around the light pulse. The asterisk indicates the over-threshold spike detected, while the gray arrows are the on-set and off-set stimulation artifacts. **c** Mean (±s.e.m.) firing rate (circles) and firing probability (squares) of SL spikes, computed across all the recorded cells ($n = 39$, 10 sweeps each) on the PDMS–photovoltaic interface. For each cell, the probability has been defined as the percentage of sweeps with at least a SL spike over the 10 consecutive trials. **d** Mean (±s.e.m.) latency (circles) and jitter (squares) of the first spike occurring in the 10 ms window after the light onset, computed across all the recorded cells ($n = 39$, 10 sweeps each) on the PDMS–photovoltaic interface. For each cell, the mean latency and jitter has been computed over the ten consecutive trials. **e**, **f** Mean (±s.e.m.) firing rate of medium (**e**) and long (**f**) latency spikes, computed across all the recorded cells ($n = 39$, ten sweeps each) on the PDMS–photovoltaic interface. In panels **c**–**f** values have been plotted up to 3 mW mm$^{-2}$, while the full profiles are shown in Supplementary Fig. 3c–f

extraction test on the murine fibroblastic L929 cells. Cell viability has been estimated via an XTT cell viability assay. Results on the prosthesis showed a 100% viability, while positive control has 0.3% viability and negative control has 100% viability (averages of three repetitions; see Certificate in Supplementary Information).

**Thermal and optical safety.** According to the thermal safety standard for active implantable medical devices (ISO 14708-1 / EN 45502-1), the maximum temperature on the surface of the

implant should not exceed 2 °C above the normal surrounding body temperature of 37 °C[31]. We measured in air the increase in temperature on the POLYRETINA surface (Fig. 9a, b) due to continuous operation for 2 h under full-field pulsed illumination (20 Hz, 10 ms, 1.22 mW mm$^{-2}$). The irradiance has been set to the maximal allowed by the LED. The mean (± s.d., $N = 4$ prostheses) thermal increase at steady state is 1.24 ± 0.29 °C, which is below the standard limit of 2 °C. We verified also that the temperature increases on the electrodes and on the polymer surface are not different (Fig. 9c, d). Anyhow, this experiment

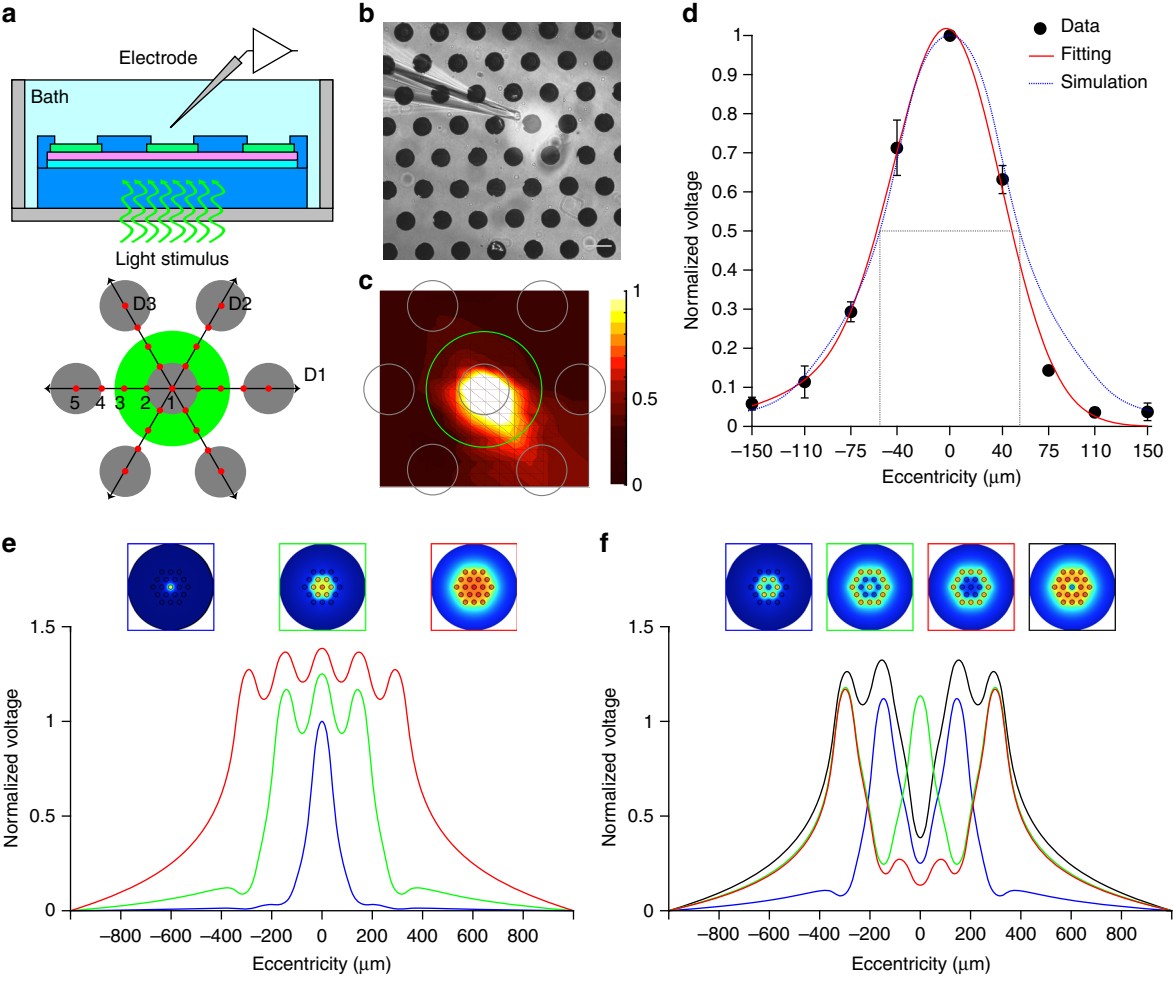

**Fig. 7** Spatial confinement of the prosthetic stimulation. **a** Sketch of the experimental setting. The green circle corresponds to the area illuminated around the central pixel. Gray circles represent the illuminated pixel and the six surrounding ones. The voltage has been measured in nine positions (red dots) for each direction (D1, D2, and D3), all cantered in the center of the illuminated pixel. **b** Picture during recordings. The light spot is visible (brighter area). The scale bar is 100 μm. **c** Voltage spreading colour map generated by interpolating the experimental measures with a triangulation-based linear interpolation. At each point ten consecutive recordings have been averaged and the voltage peaks have been normalized with respect to the value obtained in the central pixel (position 1 in **a**). The green circle is the illuminated area, while the gray circles represent the pixels. **d** Mean (±s.e.m.) normalized PV peaks from $n = 4$ pixels. For each pixel, the data from the three directions have been averaged. The red line shows a Gaussian fitting, while the blue dotted line represents the normalized voltage profile obtained by FEA simulations. The gray dotted lines show the FWHM value for the simulated profile. **e** FEA simulations for three beam sizes, normalized to the potential corresponding to the illumination of the single central pixel. **f** FEA simulations for various patterns of activation normalized to the potential corresponding to the illumination of the single central pixel

corresponds to the extreme case of projecting a constant full white frame, which is not realistic in daily operation when images will be presented as black and white. Under this condition, the average light dose is lower and therefore the related increase in temperature will be lower. In addition, the eye vitreous has a thermal conductivity about 30 times higher than air, therefore heat sinking is more efficient.

Regarding optical safety, photovoltaic prostheses are limited by retinal damage upon light exposure[32] (ANSI Z136.1 / ISO 60825 / ISO 15004). According to the standards, the maximum permissible exposure (MPE) during chronic illumination of the full POLYRETINA (equivalent to a full white frame) is controlled by the photothermal damage and equal to 328.75 μW mm$^{-2}$ (see Methods). However, photovoltaic prostheses operate with pulsed illumination. With pulses of 10 ms and duty cycle of 5, 10, or 20% (respectively for 5, 10, or 20 Hz), the MPE is increased to 6.58, 3.29, or 1.64 mW mm$^{-2}$, respectively. These values are higher

than the saturation value measured with retinal explants (1.08 mW mm$^{-2}$).

In case of POLYRETINA, the incident light is first absorbed by the P3HT:PCBM layer. The mean (±s.d., $N = 4$ prostheses) transmittance of POLYRETINA has been experimentally measured as 49.07 ± 5.25% (Supplementary Fig. 6). Therefore, only part of the incident light reaches the retina and the retinal pigmented epithelium (RPE), thus reducing the effect of retinal heating due to light absorption in the RPE. However, the light absorbed by P3HT:PCBM generates heat, that should be taken into account when evaluating the MPE. We performed FEA simulations to estimate the temperature increase in the retina upon illumination of POLYRETINA. First, we verified the temperature increase at the RPE–retina interface using the MPE obtained without POLYRETINA (328 and 1.64 mW mm$^{-2}$), respectively, for continuous and pulsed (10 ms pulses at 20 Hz) illumination. After 150 s of continuous illumination (560 nm,

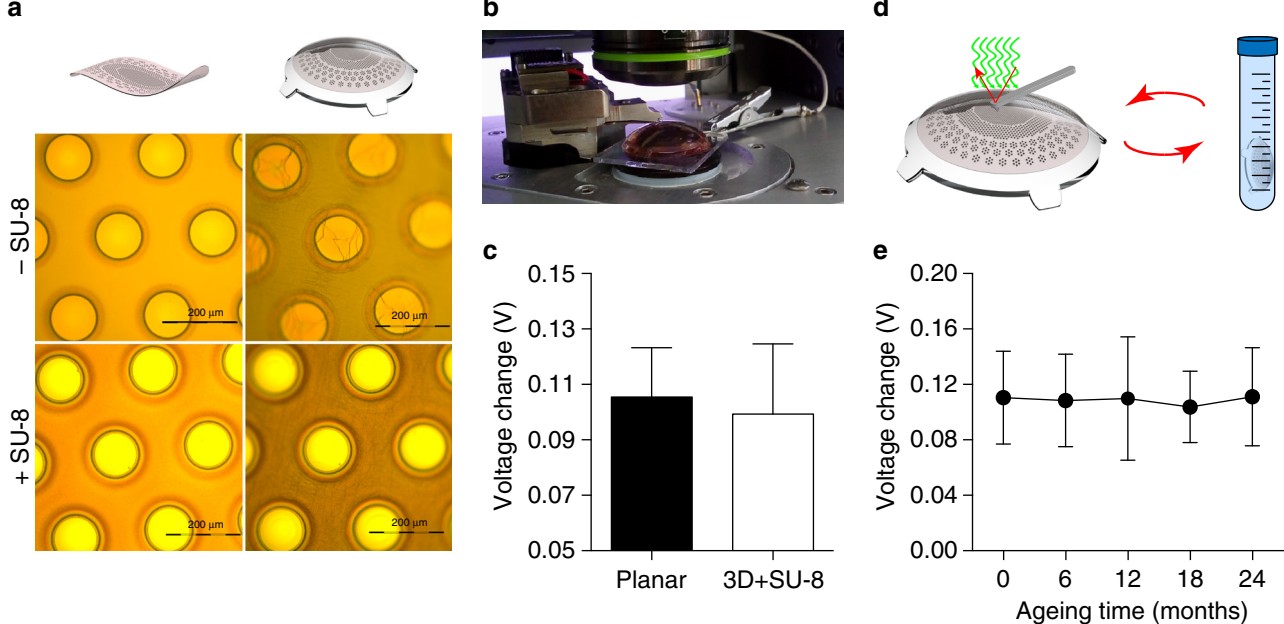

**Fig. 8** Lifetime of the retinal prosthesis. **a** Pictures of the titanium cathodes before (left column) and after (right column) bonding on the dome-shaped PDMS support. The top row is without SU-8 rigid platforms, while the bottom row is with SU-8 rigid platforms. **b** Picture of a KPFM measure on bonded prostheses integrating SU-8 rigid platforms. **c** Comparison of KPFM measures on bonded prostheses integrating SU-8 rigid platforms (99.35 ± 25.26 mV, mean ± s.d., $n = 15$; electrode diameter 80 µm) with respect to measures on PDMS-interface bonded to a planar glass substrate (105.50 ± 17.79 mV, mean ± s.d., $n = 36$; electrode diameter 100 µm). **d** Sketch of the accelerated ageing tests. KPFM measures have been performed at the beginning of the experiment, then prostheses have been immersed in saline solution at 87 °C and 100% humidity for 135 h, after that KPFM has been repeated, and on for four cycles. **e** Quantification (mean ± s.d., $N = 4$ prostheses, $n = 4$ electrodes per prosthesis) of the surface potential changes (voltage in light—voltage in dark) during accelerated ageing tests over a simulated period of 24 months (months: 0, 110.5 ± 33.53 mV; 6, 108.5 ± 33.37 mV; 12, 109.8 ± 44.59 mV; 18, 103.8 ± 25.73 mV; 24, 111.1 ± 35.48 mV)

328 µW mm$^{-2}$), the temperature increase is stable at 0.42 °C (Supplementary Fig. 7a, b). Pulsed illumination (10 ms pulses at 20 Hz, 1.64 mW mm$^{-2}$) generates temperature spikes of about 0.04 °C, oscillating around the profile corresponding to the continuous illumination (Supplementary Fig. 7c, d). This demonstrate that the scaling factor of 5 to estimate the MPE during pulsed stimulation (20% duty cycle) is correct. Continuous illumination has been used in the following simulations to reduce the computational cost. With POLYRETINA the temperature increase after 150 s of continuous illumination (560 nm, 328 µW mm$^{-2}$) is slightly reduced to 0.37 °C (Fig. 10a, b). In this case, the critical interface is the one between the retina and the prosthesis (Supplementary Fig. 8a, b) giving a slightly higher temperature increase with respect to the RPE-retina interface (0.37 vs. 0.35 °C). POLYRETINA has been simulated in direct contact with the retina because this represents the worst-case scenario. A thin space of vitreous (100 µm) between the retina and POLYRETINA reduces the temperature increase by 0.009 °C, which is negligible. Thermal damage of the retina requires a local rise in temperature higher than 10 °C[33]; the 50% of probability of retinal damage (ED50) has been previously defined for a temperature rise of 12.5 °C[31]. In our model, we estimated the ED50 with (red) and without (black) POLYRETINA (Fig. 10c). As expected the ED50 for continuous illumination is slightly higher when POLYRETINA is present (10.6 vs. 9.4 mW mm$^{-2}$), which correspond to 53 mW mm$^{-2}$ for pulsed illumination. A comparison with and without POLYRETINA showed that over the broad range of irradiances the temperature increase in the retina is reduced by 11% with POLYRETINA. Therefore, the MPE could be slightly increased to 1.84 mW mm$^{-2}$ and POLYRETINA can safely operates at 1 mW mm$^{-2}$.

## Discussion

One of the most important open questions in the field of retinal prostheses concerns how to increase both visual acuity and visual field size together. From the engineering point of view this implies to increase the density of the stimulating electrodes and enlarge the size of the prosthesis. POLYRETINA is a novel foldable and photovoltaic wide-field epiretinal prosthesis with a remarkable increase in its size (46.3 degrees) and in the number of stimulating pixels (2215) compared to other epiretinal prostheses[4,34].

Concerning visual field, POLYRETINA has the potential to cover a retinal surface corresponding to a visual angle of 46.3 degrees, which is larger than the threshold for both legal blindness (20 degrees) and adequate mobility skills (30 degrees).

Concerning spatial resolution, the presence of a continuous semiconductor layer does not represent a limitation. In organic photovoltaics, the low carrier mobility and lifetime limit the carrier–transport length to tens of nm for holes and few hundreds of nm for electrons[35]. It has been shown by another group that the PC detected at the cathode is reduced to about 10% of the maximum if the illumination spot (size 1 µm) is moved laterally by about 12 µm from the electrode edge[36]. This large decay length, beyond the simple diffusion processes, has been explained by a steady state nonlocal electric field inducing a lateral flow of the separated carriers. For this reason, an internal cross-talk between electrodes due to charge carriers generated under one electrode traveling laterally towards an adjacent electrode can be excluded (at least down to an edge-to-edge distance of about 20 µm). By measuring the voltage spread in solution together with FEA simulations we showed that the area of activation (about 100 µm) of 1 pixel is comparable to the pixel size.

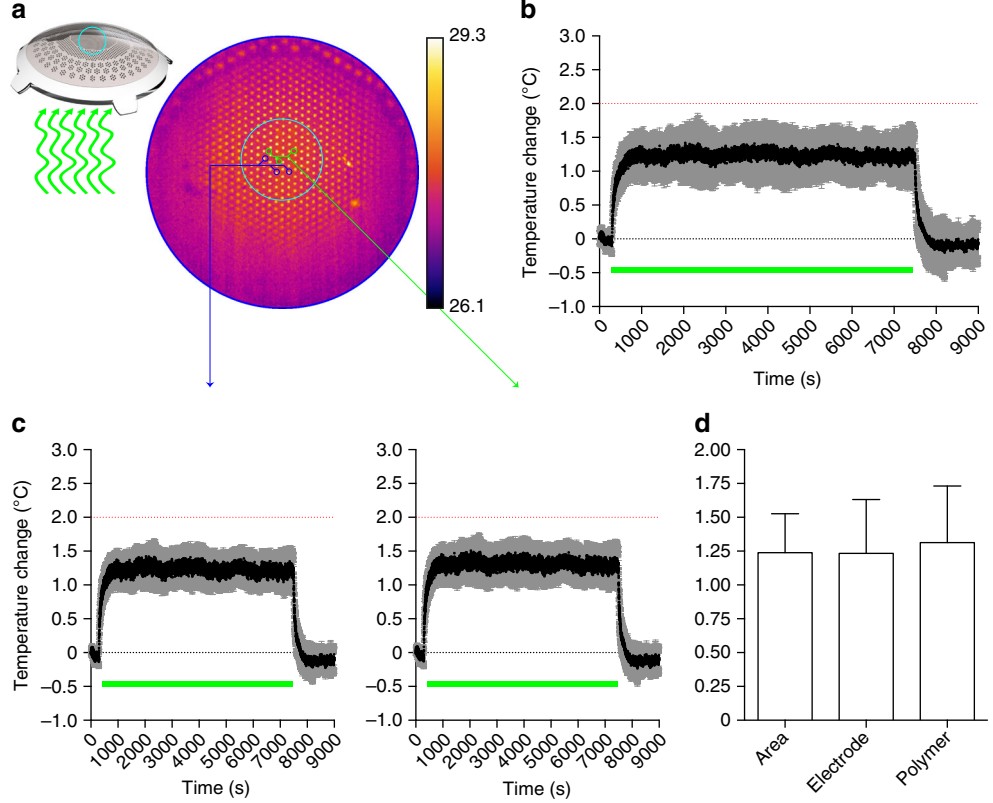

**Fig. 9** Temperature variation during operation. **a** The top surface of POLYRETINA has been imaged with a thermal camera while pulsed illumination has been provided from the bottom, as in the epiretinal configuration. The camera has been focused on the top electrodes and a ROI has been selected to measure the changes in surface temperature (cyan circle). Electrodes show higher value of baseline temperature because the metallic surface reflects part of the IR light used for the measurement. **b** Mean (±s.d.) changes in surface temperature measured in $N = 4$ prostheses. Data have been plotted has difference with respect to the baseline temperature measured for 5 min before pulsed illumination. The green bar represents the period of 2 h when light pulses have been applied (10 ms pulses, 20 Hz repetition rate, 1.22 mW mm$^{-2}$). The dotted red line represents the maximal allowed temperature increase. **c** Mean (±s.e.m.) changes in surface temperature measured on the electrodes (left, $N = 4$ prostheses) or on the polymer area (right, $N = 4$ prostheses). For each prosthesis, $n = 3$ electrodes/areas have been sampled and averaged. **d** Mean (±s.d.) changes in surface temperature in the average surface, the electrode area or the polymer area are not significantly different (1.24 ± 0.29, 1.23 ± 0.20, 1.31 ± 0.21, respectively; one-way ANOVA, $F = 0.0569$, $p = 0.9451$)

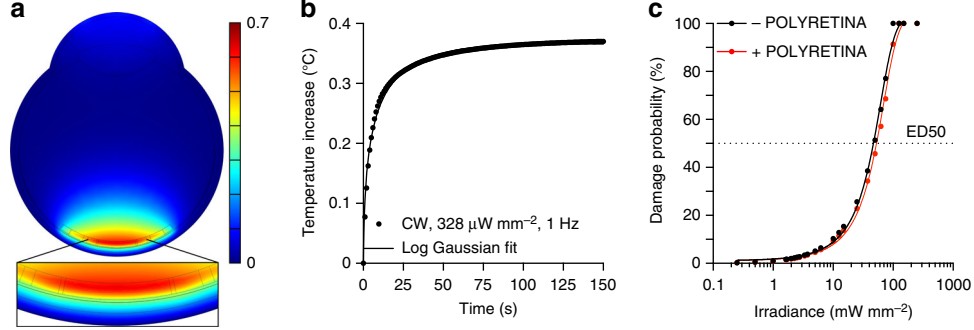

**Fig. 10** FEA simulation of thermal effects with POLYRETINA. **a** Temperature increase in the modeled eye with POLYRETINA after 150 s of continuous illumination (CW, 560 nm, 328 μW mm$^{-2}$). The insert shows a larger view of the modeled retina and POLYRETINA. **b** Time course of the temperature increase in the modeled retina during 150 s of continuous illumination (CW, 560 nm, 328 μW mm$^{-2}$). The simulation frequency has been set to 1 Hz. The solid line is the log Gaussian fit ($R^2 = 0.9934$). **c** Probability of retinal damage as a function of irradiance with (red) and without (black) POLYRETINA. ED50 corresponds to a temperature increase of 12.5 °C. The irradiance has been expressed for pulsed illumination (20% of duty cycle). The solid lines are the Sigmoidal fits ($R^2 = 0.9971$ for black and 0.9977 for red)

Concerning visual acuity, with a pitch of 150 μm the theoretical visual acuity restored by POLYRETINA is in the order of 20/600[6]; which is better than the current epiretinal prostheses (e.g., Argus II) but still below the threshold of legal blindness. However, the technology of POLYRETINA is highly scalable. Based on mechanical simulations (not shown), the pitch can be reduced down to a value of 110 μm, keeping the same electrode size (80 μm), thus approaching the theoretical value of 20/400. A further

improvement consists in reducing the size of the electrode (i.e., 60 μm) with a pitch of 80 μm, thus approaching a theoretical visual acuity of 20/300, similar to the silicon photovoltaic sub-retinal prosthesis[15]. However, these values come from theoretical computation, and therefore must be validated with proper in vivo experiments in animals and later in humans. Moreover, the reduction of the pixel size will reduce the PC generated by the interface, therefore the efficiency in stimulating RGCs should be validated again.

To be used as retinal prosthesis, POLYRETINA must operate with a stimulation rate higher that 1 Hz. The subretinal prosthesis Alpha IMS operates in a frequency range of 1 to 20 Hz (variable from patient to patient) with a pulse duration of 1–4 ms[37]. Available pulse rates in the Argus II are in the range of 3–60 Hz[38]; however, also in this case the effect of pulse rate have been reported to be very variable among subjects[39]. This suggests that, even if overall the variation in the pulse rate does not have a significant effect, an optimal pulse rate can be defined for each subject. Moreover, the recent identification of an optimal pulse duration of 25 ms per phase[40] may limit the operating range of Argus II to a theoretical limit of 20 Hz. For the silicon photo-voltaic subretinal prosthesis, the stimulation frequency is mainly limited by the discharge rate of the electrode, therefore a shunt resistor has been included to allow faster stimulations (20–40 Hz)[25] up to flicker fusion[15]. POLYRETINA shows a fast discharge of the Ti-based photovoltaic electrodes (probably due to the high shunting capacity of the P3HT:PCBM layer), and we demon-strated its functioning up to 20 Hz of stimulation rate without an additional shunting resistor. This is within the operation range of other epiretinal (e.g., Argus II) and subretinal (e.g., Alpha IMS) prostheses.

The activation of RGCs has be obtained already at 47.35 μW mm$^{-2}$ with a response saturation above 1.08 mW/mm$^2$. How-ever, recording ex vivo with retinal explants may not be repre-sentative of the complexity of retinal stimulation in vivo in humans, where the electrode-to-cell distance could be larger and increase during years of implantation[41], thus increasing the perceptual threshold. The hemispherical design is a solution to reduce the electrode-cell distance over the area of the prosthesis. Moreover, the capability of activating RGCs at low irradiance is promising in perspective of an in vivo application. In a future development, titanium/titanium nitride electrodes can be fabri-cated in order to improve the stimulation efficiency (because of their higher charge injection capacity).

The presence of SL spikes is an evidence in support of a direct activation of RGCs. On the contrary, ML and LL spikes are due to the activation of the internal retinal circuit. In literature, SL spikes are reported to be very close (i.e., 0.5–4 ms) to the stimulus[28], which is typically a sharp squared pulse. The photo voltage/cur-rent generated by POLYRETINA have a less shaper transition from 0 to the peak (in about 10 ms). This may explain why the average latency is 4.12 ± 0.07 ms and we considered as SL spikes those with a latency in the 0–10 ms window. It is known that brief (hundreds of μs) cathodic epiretinal stimulation preferentially excite RGCs, while pulses longer than 1 ms excite both RGCs and bipolar cells[42,43]. It has been recently demonstrated that the use of pulses shorter than 8 ms results in the activation of axons of passage that causes streak responses, while longer pulses results in a more focal activation[40]. Using calcium imaging techniques, authors explained this result via a shift from direct to indirect activation of RGCs. We showed by electrophysiological record-ings and pharmacological experiments that the cathodic stimu-lation provided by POLYRETINA is also indirectly activating RGCs. This represents a promising result for the in vivo trans-lation of POLYRETINA in order to obtain a focal activation. Further experiments aiming at dissecting the circuit activated by

POLYRETINA will help in defining the appropriate stimulation parameters to obtain a more focal stimulation.

Taking advantage of accelerated ageing experiments, we demonstrated that POLYRETINA preserves its optoelectronic functions unaltered for at least 2 years. More experiments and additional time points will be added to investigate the entire lifetime of the prosthesis. Last, POLYRETINA fulfils the requirements for in vitro cytotoxicity according to ISO 10993-5 and for thermal safety (ISO 14708-1/EN 45502-1).

POLYRETINA is foldable to allow implantation through a small scleral incision and it self-opens once released into the eye. Although it could operate in both epiretinal and subretinal con-ditions, it has been designed as an epiretinal prosthesis, since the implantation of a large retinal prosthesis in the subretinal space may result in an excessive damage to the remaining retinal tissue. Moreover, an epiretinal placement may allow an easier replace-ment in case of malfunction (e.g., due to ageing or detachment). The hemispherical shape has been obtained by bonding the PDMS-photovoltaic interface on a dome-shaped PDMS support with a radius of curvature of 12 mm. However, the flexibility in the fabrication process of the dome-shaped PDMS support (PDMS molding) allows the fabrication of prostheses designed to fit the real eye curvature/shape of a patient[44]. This opens up the possibility to an optimized retinal prosthesis according to per-sonal needs. Last, the shape of the prosthesis and the insertion strategy have been inspired by the widely use intra ocular lenses. With further investigations, a similar 'injection' approach could also be envisaged for POLYRETINA, simplifying even more the surgical approach. A future improvement for human use may include the removal of electrodes in correspondence of the optic nerve head and the creation of small holes within the substrate to allow metabolic exchange between the vitreous and the retina. On the functional point of view, the next step is the electro-physiological validation in vivo with large animal models, such as swine models.

## Methods
**Prosthesis micro-fabrication.** PDMS-photovoltaic interfaces were fabricated on silicon wafers. A thin sacrificial layer of poly(4-styrenesulfonic acid) solution (561223, Sigma-Aldrich) was spin-coated on the wafers (1000 rpm, 40 s) and baked (120 °C, 15 min). Degassed PDMS pre-polymer (10:1 ratio base-to-curing agent, Sylgard 184, Dow-Corning) was then spin-coated (1000 rpm, 60 s) and cured in oven (80 °C, 2 h). After surface treatment with oxygen plasma (30 W, 30 s), a 6-μm thick SU-8 (GM1060, Gersteltec) layer was spin-coated (3800 rpm, 45 s), soft-baked (130 °C, 300 s), exposed (140 mJ cm$^{-2}$, 365 nm), post-baked (90 °C, 1800 s; 60 °C, 2700 s), developed in propylene glycol monomethyl ether acetate (48443, Sigma-Aldrich) for 2 min, rinsed in isopropyl alcohol, and dried with nitrogen. After surface treatment with oxygen plasma (30 W, 30 s), a second layer of degassed PDMS pre-polymer (10:1) was spin-coated (3700 rpm, 60 s) and cured in oven (80 °C, 2 h). PEDOT:PSS (HTL Solar, Ossila) was filtered (1 μm PTFE filters) and then spin-coated (3000 rpm, 60 s) onto the $O_2$-plasma treated (30 W, 30 s) PDMS surface. Subsequent annealing (120 °C, 30 min) was performed. The pre-paration of the organic semiconductor blend was performed in a glovebox under argon atmosphere. Twenty milligrams of P3HT (698997, Sigma Aldrich) and 20 mg of PCBM (M111, Ossila) were dissolved in 1 ml of anhydrous chlorobenzene each and let stirring overnight at 70 °C. The solutions were then filtered (0.45 μm PTFE filters) and blended [1:1 v:v]. The P3HT:PCBM blend was then spin-coated at 1000 rpm for 60 s. Titanium cathodes were deposited by DC sputtering through a shadow mask aligned with the SU-8 pattern. After surface treatment with oxygen plasma (30 W, 15 s), the encapsulation layer of degassed PDMS pre-polymer (5:1 ratio) was spin-coated (4000 rpm, 60 s) and cured in oven (80 °C, 2 h). To expose the cathodes, photolithography and PDMS dry etching were performed. The wafers were then placed in deionized water to allow dissolution of the sacrificial layer and the release of the PDMS-photovoltaic interfaces. The floating membranes were finally collected and dried in air. The dome-shaped PDMS supports were fabricated using a milled PMMA mold, filled with PDMS pre-polymer (10:1), which was then degassed and cured in oven (80 °C, 2 h). The supports were released from the molding parts and perforated with a hole-puncher (330 μm in diameter) at the locations dedicated to the insertion of retinal tacks. The released PDMS-photovoltaic interfaces were clamped between two O-rings and, together with the PDMS supports, were exposed to oxygen plasma (30 W, 30 s). The activated PDMS surfaces were put in contact and allowed to uniformly bond thanks to radial

stretching of the fixed membrane. The excessive PDMS used to clamp the array was removed by laser cutting.

**Chips micro-fabrication.** Chips for KPFM and PC/PV measurements were fabricated on $20 \times 24$ mm$^2$ glass substrates (2947–75 × 50, Corning Incorporated). Before micro-fabrication, glass chips were cleaned by ultra-sonication in deionized water, acetone and isopropyl alcohol for 15 min each and then dried with nitrogen. ITO (200 nm) was deposited on glass chips by RF sputtering. PEDOT:PSS (HTL Solar, Ossila) was filtered (1 μm PTFE filters) then spin-coated at 3000 rpm for 60 s on each chip. Subsequent annealing at 120 °C for 30 min was performed. The preparation of the organic semiconductor blend was performed as described before. The P3HT:PCBM blend was then spin-coated at 1000 rpm for 60 s on each chip. Aluminum cathodes were deposited by thermal evaporation using a shadow mask; titanium cathodes were deposited by DC sputtering using a shadow mask. When present, degassed PDMS pre-polymer (10:1) was deposited on the glass substrate by spin-coating (1000 rpm, 60 s) and then cured in oven (80 °C, 2 h).

**Kelvin probe force microscopy.** KPFM characterization was performed in ambient conditions with an Asylum Research Cypher S microscope using PtIr coated tips (AC240TM, Asylum Research, Oxford Instrument) in surface potential imaging mode. To measure the surface potential variation, KPFM images were collected by repetitively scanning a single line of 100 nm under dark and illumination conditions. The white LED of the microscope positioned above the tip and sample was acting as light source and it was manually turned 100 % off and 100 % on to reach the desired conditions. KPFM images were analyzed using Gwyddion 2.36 software. For each image, the average surface potential variation value was obtained by subtracting the surface potential under illumination to the one under dark (voltage in light–voltage in dark).

**Accelerated ageing tests.** Accelerated ageing was performed in a dry oven set to 87 °C. Samples were immersed in physiological saline solution (0.9 % NaCl, pH 7.4) within a sealed 50-ml falcon tube. Under this condition, the acceleration factor was 32[45,46]. KPFM measures were obtained before starting the ageing protocol and at several time points during ageing. Each accelerated ageing session between KPFM measures lasted for 135 h, corresponding to 6 months. Before KPFM, samples were removed from the sealed falcon tube, rinsed with deionized water, and dried under nitrogen flow.

**Measure of PV and PC.** In this experiment, the photovoltaic interface has been fabricated directly on glass (without PDMS) to avoid breaking the connecting lines from the electrode to the pad when contacted. The titanium electrodes have been fabricated with a diameter of 100 μm; however, when evaluating the PC density generated by the interface, also the area of the connecting line exposed to light has been considered. A plastic reservoir was attached to the chip using PDMS as adhesive. Chips were placed on a holder, and each pad was sequentially contacted. Silver paste was used to improve the electrical contact. An Ag/AgCl pellet immersed in physiological saline solution (NaCl 0.9 %) was used as reference electrode. Light pulses were delivered by a 565-nm Green LED (Thorlabs, M565L3-C5) focused at the sample level. PV was measured using a voltage amplifier (DL-Instruments, 1201; gain 20, band DC-3000 Hz) and PC using a current amplifier (DL-Instruments, 1212; gain $10^{-6}$ A/V). Data sampling (16 kHz) and instrument synchronization were obtained via a DAQ board (PCIe-6321, National Instruments) and a custom-made software. Data analysis was performed in Matlab (Mathworks). Due to a limitation in the acquisition system, long pulse trains (Fig. 5e) have been delivered in packages of 20 pulses at 20 Hz (total of 1 s), while each package was separated by 1 s needed by the system to save data before starting the next package.

**Electrophysiology.** Experiments were conducted under the animal authorizations VD3055 and GE3717. Retinas were explanted in normal light conditions from mice sacrificed by injection of Sodium Pentobarbital (150 mg kg$^{-1}$). After eye enucleation, retinas were dissected in carboxygenated (95% O$_2$ and 5% CO$_2$) Ames' medium (A1420, Sigma-Aldrich) and transferred to the microscope stage for recordings. In the experiment with synaptic blockers, Ames' medium was supplemented with DL-AP4 (250 μM l$^{-1}$, No. 0101, Tocris Bioscience), DL-AP5 (50 μM l$^{-1}$, No. 0105, Tocris Bioscience), DNQX (10 μM l$^{-1}$, No. 0189, Tocris Bioscience), Carbenoxolone (100 μM l$^{-1}$, No. 3096, Tocris Bioscience). Retinas were placed mimicking an epiretinal configuration, therefore with RGCs facing the substrate (bare PDMS or prosthesis). On the prosthesis, retinas were layered in the central part of the array with electrodes of 80 μm in diameter and 150 μm pitch. Recordings were performed in dim light at 32 °C with a sharp metal electrode (PTM23BO5KT, World Precision Instruments), amplified (Model 3000, A-M System), filtered (300–3000 Hz), and digitalized at 30 kHz (Micro1401–3, CED Ltd.). Illumination was carried out on a Nikon Ti-E inverted microscope (Nikon Instruments) by the Spectra X system (Emission filter 560/32, Lumencor). The microscope was equipped with a dichroic filter (FF875-Di01–25 × 36, Semrock) and a 10× (diameter of the illumination spot 2.2 mm; CFI Plan Apochromat Lambda) objective. The stimulation protocol consisted in a repetition of ten pulses at 1 Hz for each irradiance; irradiance was increased sequentially: LED at 0% (0

μW mm$^{-2}$), 2% + ND4 (47.38 μW mm$^{-2}$), 3% + ND4 (107.91 μW mm$^{-2}$), 2% (189.50 μW mm$^{-2}$), 3% (421.12 μW mm$^{-2}$), 3% (815.92 μW mm$^{-2}$), 5% (1081.75 mW mm$^{-2}$), 10% (2.81 mW mm$^{-2}$), 20% (5.89 mW mm$^{-2}$), 40% (11.98 mW mm$^{-2}$), 60% (17.92 mW mm$^{-2}$), 80% (23.56 mW mm$^{-2}$), and 100% (29.08 mW mm$^{-2}$). Spike detection and sorting were performed by threshold detection using the Matlab-based algorithm Wave_clus[47] and further data processed in Matlab (Mathworks). The threshold for spike detection has been defined as 3.7 times the standard deviation of the background noise. The minimum refractory period between spikes of the same class was set to 1.4 ms. To ensure the rejection of artifacts, an exclusion period of ± 1 ms around light onset and offset was applied. However, the spikes in the first 10 ms after the light onset (SL) have been manually verified. PSTHs for each condition of illumination were computed discretizing and averaging spike raster obtained over ten stimulations repetitions into bins of 10 ms. Spikes were sorted from individual PSTHs and classified according to their timing after light onset (cyan bars in Supplementary Fig. 3a) in SL (<10 ms), ML (from 40 to 120 ms), and LL (from 150 to 350 ms)[28]. Firing rates in the three groups were measured as follow. For SL spikes the first bin (10 ms) after the pulse was used. For ML spikes 3 bins (30 ms) in the defined time range, centered in the highest bin, were used. For LL spikes 5 bins (50 ms) in the defined time range, centered in the highest bin, were used.

**pH measurements.** Experiments have been performed in phosphate-buffered saline at room temperature. Illumination was carried out on a Nikon Ti-E inverted microscope (Nikon Instruments) by the Spectra X system (Emission filter 560/32, Lumencor). The microscope was equipped with a dichroic filter (FF875-Di01–25 × 36, Semrock) and a 10× (diameter of the illumination spot 2.2 mm; CFI Plan Apochromat Lambda) objective. Light pulses of 10 ms where delivered at 20 Hz for 1 h with an irradiance of 3.4 mW mm$^{-2}$. pH was measured with a microelectrode (tip diameter of 200 μm) with internal reference (pH-200C, Unisense). Data were sampled at 1 H.

**Spatial selectivity measures and modeling.** Measures of the voltage spread have been performed in Ames' medium (A1420, Sigma-Aldrich) at 32 °C with a glass micropipette (tip diameter about 15 μm). Data were amplified (Model 3000, A-M System), filtered (DC-1000 Hz), and digitalized at 30 kHz (Micro1401–3, CED Ltd.). Illumination was carried out on a Nikon Ti-E inverted microscope (Nikon Instruments) by the Spectra X system (Emission filter 560/32, Lumencor). The microscope was equipped with a dichroic filter (FF875-Di01–25 × 36, Semrock) and a 10 × objective. A pin-hole was used to limit the spot diameter to about 150–170 μm. After alignment of the illumination spot on a target pixel of the central area of POLYRETINA, ten pulses of 10 ms were delivered at 1 Hz with an irradiance of 29.07 mW mm$^{-2}$. The resulting voltage has been measured at nine positions in three directions around the illuminated pixel. Data analysis was conducted in Matlab (Mathworks). Voltage peaks above noise level (mean noise threshold 6.2 μV) have been detected and their amplitude normalized respect to the central pixel value. Simulations were performed in COMSOL Multiphysics 5.2, with a stationary electric currents study. The titanium cathodes were set at 0.1 V, while PEDOT:PSS was put at 0 V. The ground was situated at the bath top and lateral walls that were placed 2 mm and 1 mm away from the central pixel, respectively (cylindrical geometry). Line plots shown in the results were taken at a distance of 20 μm from the titanium surface. For each material, the conductivity (S m$^{-1}$) and relative permittivity is listed: titanium ($2.6 \times 10^6$/1), P3HT:PCBM (0.1/3.4), PEDOT:PSS (30/3), Saline (1/80), PDMS ($2 \times 10^{-14}$/2.75).

**Optical safety.** Retinal damage upon light exposure can occur because of three main factors: photo-thermal damage, photo-chemical damage, and thermo-acoustic damage[31]. The first one is related to retinal heating upon light absorption by the melanin in the RPE. The second one occurs at short wavelengths (less than 600 nm) and for exposures longer that 1 s. The latter occurs for short pulses (less than 1 ns) and is associated with nonlinear photo-mechanical effects. POLY-RETINA functions with 10 ms green light pulses; therefore, this limit could be controlled by the photo-thermal or photo-chemical damage. According to the ANSI Z136.1 Standard[32], the MPE allowed for ophthalmic applications can be calculated (in W) according to equation (1) for photo-thermal damage (MPE$_T$) and equation (2) for photo-chemical damage (MPE$_C$). Those equations are valid for $\lambda$ = 560 nm and $\alpha$ = 808.12 mrad (Supplementary Fig. 1c).

$$\text{MPE}_T = 6.93 \times 10^{-5} C_E C_T \frac{1}{P}$$

$$\text{with} \begin{cases} C_E = 6.67 \times 10^{-3} \alpha^2 \text{ for } \alpha > 100 \text{mrad} \\ C_T = 1 \text{ for } 400 < \lambda < 700 \\ P = 5.44 \text{ for } 400 < \lambda < 600 \text{ and } t \geq 0.7s \end{cases} \quad (1)$$

$$\text{MPE}_C = 5.56 \times 10^{-10} C_B \alpha^2 \text{ with } C_B = 10^{0.02(\lambda - 450)} \quad (2)$$

MPE$_T$ results in 47.41 mW, which corresponds to 328.75 μW mm$^{-2}$ for an exposed area of 144.22 mm$^2$. MPE$_C$ results in 57.55 mW, which corresponds to 399.02 μW mm$^{-2}$.

**Thermal measurements.** Measures have been performed with a thermal camera (FLIR A325sc Infrared Camera, FLIR Systems, Inc.) focused on the top surface of the POLYRETINA prosthesis. Images have been acquired at 1 frame per second. Light pulses (10 ms, 20 Hz, 1.22 mW mm$^{-2}$) were delivered by a 565-nm Green LED (Thorlabs, M565L3-C5) focused at the sample level.

**Thermal modeling.** Simulations were performed in COMSOL Multiphysics 5.2 with the Bioheat module for the heat transfer equation and the General PDE module for the Beer–Lambert light propagation. Illumination has been modeled as a uniform beam with a diameter of 13 mm. The eye is a 2D axi-symmetric model composed of several spheres representing each domain (Supplementary Fig. 9). A total of 8 domains (Cornea, Aqueous Humor, Lens, Vitreous Humor, Retina, RPE, Choroid and Sclera) are defined in the model, with the parameters listed in Supplementary Table 1. POLYRETINA was modeled as a single composite material, with volume averaged properties of PDMS, Pedot:PSS, P3HT:PCBM and Titanium (Supplementary Table 2). It was simplified into 5 domains with homogeneous properties: the center, the first ring, the second ring, the domains where no titanium is present, and PDMS only (Supplementary Fig. 9). A volume average has been performed on each of this domain, to obtain the parameters for the aggregated material. To account for the non-uniform distribution of titanium, the fraction area of titanium was considered. To validate the parameters of the aggregated model, a simulation has been performed with POLYRETINA in air exposed to continuous illumination (560 nm, 244 µW mm$^{-2}$) corresponding to pulsed illumination of 1.22 mW mm$^{-2}$. The heat losses at the prosthesis interface-air were radiative (emissivity = 0.9) and convective (heat transfer coefficient = 38.5 W m$^{-2}$ K$^{-1}$). In agreement with our experimental results, the average transmittance was measured to be 51.67% (49.07% in Supplementary Fig. 6) and the steady-state temperature increase was 1.25 °C (1.24 °C in Fig. 9).

**In vitro cytotoxicity test.** The study validation was performed by an accredited company (Medistri SA). The test was conducted according to the requirement of ISO 10993-5: Biological Evaluation of Medical Devices, in vitro cytotoxicity test; ISO 10993-12: Test article preparation and reference materials; USP 35-NF30 (87): Biological Reactivity test, invitro; Medistri internal procedure WI 47 and WI 56. Prostheses were sterilized with EtO prior the test. The test on extraction was performed with two retinal prostheses for a total surface area of 3.54 cm$^2$, with a ratio of the product to extraction vehicle of 3 cm$^2$ ml$^{-1}$. Extraction vehicle was Eagle's Minimum Essential Medium supplemented with fetal bovine serum, penicillin–streptomycin, amphotericin B, and L-glutamine. The extraction was performed for 24 h at 37 °C. The extract was added on triplicate cultures wells containing a sub-confluent L929 cell monolayer (1:1 dilution). The test samples and the control wells were incubated at 37 °C in 5 % CO$_2$ for 24 h. Following incubation, the cell cultures were examined for quantitative cytotoxic evaluation. 50 µl per well of XTT reagent was added to the cells then incubated at 37 °C in 5 % CO$_2$ for further 3–5 h. An aliquot of 100 µl was then transferred from each well into the corresponding wells of a new plate and the optical density was measured at 450 nm.

**Surgical implantation.** Plastic eye models (Eyelabinnovations, Austria) and enucleated pig eyes were used. First three 23-gauge transconjunctival valved canulas (DORC, Zuidland, The Netherlands) were inserted into the eye at 4 mm from the limbus at the following positions: nasal superior, temporal superior and temporal inferior. A balanced salt solution infusion was hooked up to the eye to maintain a constant intraocular pressure through one of the cannulas. A 6.5-mm long incision was then performed using a 15° scalpel. The implant was folded using special forceps and then inserted through the incision into the posterior cavity. Once inside the eye the forceps grip was released and the implant could unfold. Using a light pipe and an intraocular 23-gauge forceps inserted through the other two cannulas the implant was then manipulated and fixed in epiretinal configurations.

**Statistical analysis and graphical representation.** Statistical analysis and graphical representation were performed with Prism (GraphPad Software Inc.). The normality test (D'Agostino & Pearson omnibus normality test) was performed in each dataset to justify the use of a parametric or non-parametric test. In each figure $p$-values were represented as: $*p < 0.05$, $**p < 0.01$, $***p < 0.001$, and $****p < 0.0001$. Data are reported as mean ± s.e.m. or mean ± s.d., $n$ is used to identify the number of electrodes or cells used; $N$ is used to identify the number of devices or animals.

**Data availability.** The authors declare that all other relevant data supporting the findings of the study are available in this article and in its Supplementary Information file. Access to our raw data can be obtained from the corresponding author upon reasonable request.

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

## Acknowledgements

We would like to acknowledge the EPFL center of micronanotechnology for the support. Prof. Matthias Lütolf for having reviewed our manuscript and Prof. Stéphanie Lacour for her advices. This work has been supported by École polytechnique fédérale de Lausanne, Medtronic, European Commission (EU project 701632), Fondation Pierre Mercier pour la science, and Velux Stiftung (Project 1102).

## Author contributions

L.F. fabricated the devices and performed/analyzed KPFM, PV/PC, temperature, and accelerated ageing tests. M.J.I.A.L. designed, fabricated, and characterized the devices and the retinal prostheses; she performed/analyzed PV/PC measures and electrical simulations. N.A.L.C. performed/analyzed pH, voltage spreading, and electrophysiological experiments. M.B. performed/analyzed PV and PC measures. S.C.A.G. performed thermal simulations. T.J.W. performed the simulated surgeries. P.V. performed the simulated surgeries. K.S. participated in the fabrication and characterization of the prostheses. D.G. designed and led the entire study, validate the data analysis, and wrote the manuscript. All the authors read and accepted the manuscript.

## Additional information

**Competing interests:** The authors declare no competing interests.

