## [Peer Review File · Nature Communications]

Reviewers' comments:

Reviewer #1 (Remarks to the Author):

This manuscript describes a new type of photovoltaic epiretinal prosthesis. The authors developed, fabricated and tested ex-vivo those implants with high visual field and high pixel count. The use of photovoltaic polymers allows wireless delivery of power and information through the pupil as well flexibility of the device for easier implantation. They additionally provide some biocompatibility data and claim thermal safety in the range of intensities needed for stimulation.

This study is original and could have a great impact in the field of retinal prosthetics but it still suffers from some lack of clarity regarding the electric field generated by the prosthesis, charge balancing, poor statistical power in the ex-vivo measurements and some mistakes in terms of thermal safety.

I try to suggest additional experiments that would greatly strengthen the study and provide convincing evidence of the usefulness of the device.

More specifically:

- there is no insulation layer between each electrode of the device. Can you detail how you expect the voltage to spread to neighboring pixels from a spatially limited light stimulation? If stimulation occur at the neighboring pixel, how can you claim any spatial selectivity?

It should be easy to measure the lateral spread of voltage with a micropipette above the device with a local light stimulation (like 200um diameter spot).

- Although low voltage and below the window for electrode damage, the monophasic nature of the pulse will trigger pH modifications and reactions in the tissue. Can you please discuss this issue and quantify it with pH imaging techniques for example?

- 1000 or 10.000 pulses to look at evolution of the pulse shape is vey low (corresponding to 1000s of use for 10Hz stimulation).

It is also not clear why 0.5Hz is required as modulation frequency. We can expect very different results from a continuous stimulation at 10Hz compared to alternative stimulation at 0.5Hz envelope. Patients would likely use those devices for a few hours per day which at least means 1 order of magnitude more in continuous use and 2 orders more for chronic use over the years. Can you please provide the evolution of the pulse shape and amplitude over at least 1 day of continuous stimulation (10Hz, 10ms, 1mW/mm² for example - full implant illumination)?

- Figure 3c : why not plotting the PC and PV along with the irradiance instead of using colorbars? It would provide a visual on the linearity of the charge injection capabilities.

- Ex-vivo study:

The number of retinas and cell used is extremelly limited. Only 8 cells for the most interesting experiment (the text mentions only 4 cells but the figure states 8 by the way)!! (figure 5). This is NOT sufficient. It also explains why there are discrepancies between figure 4 and 5.

The maximum value for the spiking rate for low latency spikes at 1mW/mm² on figure 5 is the same as the one for 10000uJ/mm² on figure 4!!

This clearly shows that the number of experiments is not sufficient and there are big mismatch in the datasets. Both figure 4 and 5 should reflect the same results and be integrated in a single figure and I suspect they were split because of those discrepancies.

I would suggest removing the data from the high power stimulation which is thermally non sustainable anyways and do more experiments at lower irradiances (at least 100 cells).

- Finally about thermal safety:

The assumptions for calculation of safety are wrong.

- the maximum permissible exposure is calculated for a visual angle of 5deg whereas the device itself is designed to provide high visual field (42deg).

- the energy absorbed by the implant does not magically disappear. It is converted in heat with less efficient cooling compared to the RPE/retina due to lack of perfusion.

- In those conditions, 1mW/mm² is close or above the thermal safety limits.

Please integrate in your assumption the real beam size powering the chip (42deg), the fact that the implant absorbs some energy and is actually heated.

Overall, this paper describes a nice technology but does not provide enough data to demonstrate its usefulness and efficiency.

Reviewer #2 (Remarks to the Author):

Dear Authors,

Congratulation to your work on a wide field photovoltaic prosthesis to cure receptor degenerations. Your device holds the promise to overcome certain limitations of devices which are already on the market.

Before I can give my recommendation to publish your manuscript I would like to ask you if you can exactly describe how it will function under real life conditions. You described how defined light stimuli can activate pulse stimulation via the electrodes in your stimulator. You described stimulus frequencies of 1- 10 Hz. How does this fit to a real life condition where images onto the retina are faster. Does your system uses ambient light or is it necessary to wear an external device for projecting a light beam to the desired spot on the implant? So please describe the system architecture.

Some other issues are listed here. I wish that you can comment on each of these points.

Page 2, first passage:

The definition of blindness depends on the country where you are. So please indicate for what country this definition is valid.

These diseases are rare diseases. Degenerative diseases such as macular degeneration, glaucoma or diabetes are much more frequent reasons for blindness. The statement could be misunderstood. It sounds that it is easy to identify patients suitable for the implantation. In reality it is extremely difficult to find appropriate patients for an implantation.

Page 2, line 25:

The stiffness depends on the thickness. Using very thin polyimide e.g. could result in a very flexible substrate.

Page 3, line 23:

The inner surface of the eye does not have a simple spherical shape. The retinal surface could vary individually with many different local radii.

Page 4, line 8:

PDMS may also have a disadvantage in terms of its failure to seal electronics from water on the long run.

Page 5, line 1:

How close is this model to the real situation? Can you provide images showing the close approximation of the device with the retina?

Page 5, line 2-4:

Do you already have any experiences with the implantation in animals? The implantation of large devices through a large opening may result in introducing the device during a hypotensive phase which makes the implantation procedure relatively uncontrolled. Plastic models are not suitable to simulate this situation because these models do not collapse.

Page 7, line 5:

With such stimulus frequencies moving objects can not be sufficiently perceived. Usually image frequencies of 24-25 Hz are used to obtain perception of moving objects. Simultaneously the stimulation of ganglion cells may require much higher stimulus frequencies with very short pulses. The system will lose its flexibility if it is limited to a low stimulation rate.

Page 10, line 18:

Again, PDMS is not a safe seal.

Page 11, line 8-10:

It is true that according to ISO 10993 the mentioned tests have to be performed. However, it would be great to see that the device can be safely placed in eyes of mid size animals and only then (clinical exams, histology, evidence for cortical activation) one can decide if the large device is suitable for implantation or not.

Sincerely

Reviewer #3 (Remarks to the Author):

Ferlauto, Airaghi Leccardi, Chenais et al. present a new design for a foldable photovoltaic epiretinal prosthesis. Their device is flexible and can consequently be implanted through a relatively small scleral incision. The implant is curved so as to remain in tight contact with the retina over a large area, theoretically enabling electrical activation of the retina over a wide field of view in blind patients (42 degrees, significantly larger than existing implants). The authors detail manufacturing techniques and characterize their implant in-vitro using test devices and ex-vivo using a mouse model of retinal

degeneration.

While the manuscript is well written and has the potential to appeal to the wider medical implants community where flexible electronics are becoming increasingly relevant, in addition to the retinal prosthesis community, some claims made by the authors need better substantiation, especially the claims regarding high visual acuity and high fidelity in retinal stimulation.

Regarding high visual acuity: at the abstract level and in the discussion (p.11 l.14-16), the authors state that their device represents a solution to improve the visual acuity of retinal prostheses, by "increasing the density of stimulation sites". However, in the densest areas of the implant (5-mm diameter central area), the device only embeds 967 pixels, which should translate to an electrode pitch of roughly 130 μ m, assuming uniform coverage of the central area (it would be helpful to know the actual manufactured electrode pitch). The electrode density of this device must then be four times lower than the current densest devices already described in the literature (Alpha IMS subretinal implant by Retina Implant AG, which should be mentioned in the introduction, pixel pitch on the order of 60 μ m, or Stanford photovoltaic implant, cited in refs 12 and 13, pixel pitch on the order of 65 μ m).

From Nyquist sampling considerations, the best visual acuity this device can hope to achieve is on the order of 20/500, still below the legal limit for blindness (20/200 USA, 20/400 WHO), and a figure that's already been achieved by some of the patients of the Alpha IMS patient cohort. The language in the text should therefore be moderated accordingly.

Additionally, recent evidence seems to indicate that the number of pixels required to perform tasks such as recognition of daily life objects is closer to a few thousands than the few hundreds cited in the text (figure from p.2, l.28, refs 10, 11): see the work of Jung et al., *Vision Res.* 2015 Jun; 111(0 0): 182–196. It therefore seems that the electrode density needs to be significantly increased before the device can claim to hope to restore vision with high visual acuity. That being said, the photovoltaic design shown here appears to be highly scalable, so it could be sufficient to discuss what it would take to achieve a higher electrode density.

Increased electrode density does not however necessarily translate into improved visual acuity, so more confidence that the device behavior is currently limited by its pixel pitch and not flaws in design is required. For example, cross-talk between neighboring electrodes because of inadequate grounding, or because of diffusion of photovoltaically-induced carriers between one pixel and the next could all contribute to a degradation of the implant performance in practical situations.

From the design shown in Supplementary Figure S1, it is unclear how one could achieve focal stimulation with the current device, for the two following reasons. (1) Assuming that tight contact is maintained between the retina and the implant, currents would then get channeled laterally around the device back to the PEDOT anode, leading to strong cross-talk between neighboring pixels. (2) With a single layer of P3HT:PCBM, what's to prevent photocarriers from diffusing away from the target pixel and actually be injected at the site of the neighboring electrode? Silicon photovoltaic implants (refs 12, 13) include physical gaps or silicon oxide trenches between pixels to prevent diffusion of the photocarriers.

Some modeling work, or current mapping similar to what was done by Flores et al., *Journal of Neural Engineering* 13: 036010 would give confidence that the device could theoretically be able to sustain high-resolution percepts. The modeling work would ideally clarify to what extent cross-talk should be expected between adjacent pixels (from carrier diffusion and the grounding scheme), and whether small pixels (ideally at least as small or smaller than the Alpha IMS and Stanford photovoltaic implant) can provide enough charge to stimulate the retina despite a decreased light-sensitive area, given a realistic estimated distance between the implant and the retina. Published current values in the

literature can be inadequate, especially when electrode sizes vary, and distance between cells and electrodes are not the same as those of the original study.

Regarding fidelity of retinal stimulation: in terms of physiological testing, some choices of stimulation parameters in the ex-vivo evaluation are surprising.

It seems unnecessary to show retinal responses for stimulation irradiances above around 10 mW/mm² when safety considerations for chronic exposure will limit the available power to 7.77 mW/mm² or lower, depending on the pulse rate (e.g. Supplementary Figures 2 and 3). The measurements taken between 10 and 100 mW/mm² were taken with repeated stimuli, so they depart from the assumptions made in the single pulse exposure calculation to show safety on p.9 (it's good to know though that single pulse limits are so high, so I am not suggesting removing this derivation).

In places, thresholds are reported in terms of energy deposited onto the implant (e.g. p.8, but also Supplementary Figure 6). It is useful to know both what pulse width and what irradiance were used, and not just energy, as stimulation thresholds depend jointly these two parameters through the strength-duration relationship for the stimulated tissue.

In the discussion, the authors mention (p.12, l.20-23) the high fidelity in retinal stimulation of their implant. However, reproducing the retinal code with high fidelity requires stimulation at frequencies of up to several hundred Hz (see for example Fried et al., *J Neurophysiol* 95: 970–978, 2006), which does not seem attainable with a photovoltaic implant that passively discharges between pulses. Boinagrov et al., *IEEE Trans. Biomed. Circuits and Systems* 10(1): 85-97, 2016 suggests that a shunt resistor is required to help discharge stimulation electrodes between pulses for a photovoltaic devices activated at a few 10s of Hz, which is missing in this implant.

Additionally, l.23-25, the authors mention that ML and LL spikes contain signals coming from activation of passing axons. This is incorrect: axonal activation also has a short latency, see Grosberg et al., *J Neurophy* DOI: 10.1152/jn.00750.2016 for a detailed description of electrical signatures of axonal activation. Independently of this issue, the evidence in the literature currently points to benefits of long pulses being delivered epiretinally to target the inner retina and bypass the RGC layer. For example, Weitz et al. in *Sci Transl Med.* 2015 Dec 16;7(318):318ra203 show they can improve the resolution of an epiretinal array by increasing stimulus duration. This study should be cited in this section of the discussion section (p.12 l.20 and after). It appears to me that the authors would want to optimize activation of the INL without activation of the RGC and axon layer instead of trying to selectively activate the RGCs only, unless they can come up with another method for bypassing axonal activation (which seems hard with short pulses, see Grosberg et al. 2017).

We would like to thank all the Reviewers for the careful revision and the interesting comments. We found them very useful to improve our manuscript. Answers have been highlighted in blue.

Reviewers' comments:

Reviewer #1 (Remarks to the Author):

This manuscript describes a new type of photovoltaic epiretinal prosthesis. The authors developed, fabricated and tested ex-vivo those implants with high visual field and high pixel count. The use of photovoltaic polymers allows wireless delivery of power and information through the pupil as well flexibility of the device for easier implantation. They additionally provide some biocompatibility data and claim thermal safety in the range of intensities needed for stimulation. This study is original and could have a great impact in the field of retinal prosthetics but it still suffers from some lack of clarity regarding the electric field generated by the prosthesis, charge balancing, poor statistical power in the ex-vivo measurements and some mistakes in terms of thermal safety. I try to suggest additional experiments that would greatly strengthen the study and provide convincing evidence of the usefulness of the device.

More specifically:

1. There is no insulation layer between each electrode of the device. Can you detail how you expect the voltage to spread to neighboring pixels from a spatially limited light stimulation? If stimulation occur at the neighboring pixel, how can you claim any spatial selectivity? It should be easy to measure the lateral spread of voltage with a micropipette above the device with a local light stimulation (like 200um diameter spot).

We would like to clarify that even if the layer of P3HT:PCBM (semiconductor) is continuous in the device, photo-generated charges cannot travel laterally for large distances and cause interference in adjacent titanium electrodes. This would certainly be an important factor in traditional inorganic photovoltaics (e.g. silicon) where free carriers are generated directly upon light absorption and diffuse long distances (even up to mm) in the film. This is why, in silicon devices (e.g. Silicon Retina developed in Stanford by Prof. Palanker) physical gaps or silicon oxide trenches are necessary between pixels. However, in the organic semiconductor P3HT:PCBM blend used in this work, it is well known that the active area of the device is defined by the area of the cathode (usually Al, but Ti is used in this case) due to the poor (hopping based) transport of free charge carriers. Indeed, in organic photovoltaics, the low carrier mobility and lifetime limit the carrier-transport length to tens of nm for holes and few hundreds of nm for electrons (doi:10.1103/PhysRevB.82.205325). Experimentally it has been shown by another group that the photocurrent detected at the cathode is reduced to ~10% of the maximum if the illumination spot (size 1 μm) is moved laterally by ~12 μm from the electrode edge (doi:10.1063/1.2998540). This large decay length, beyond the simple diffusion processes, has been explained by a steady state nonlocal electric-field inducing a lateral flow of the separated carriers. For this reason, an internal cross-talk between electrodes due to charge carriers generated under one electrode traveling laterally towards an adjacent electrode can be excluded (at least down to a edge-to-edge electrode distance of ~20 μm). This is why POLYRETINA does not require trenches between pixels. This explanation has been introduced in the new manuscript at Page 10 (line 23). Following the suggestion of both Reviewer 1 and 3, we also performed the suggested experiment coupled by FEA simulations. These additional evidences shows that it is possible to claim a spatially selective stimulation with the current device. These new evidences are included in Figure 7 and described at Page 8 (line 6).

2. Although low voltage and below the window for electrode damage, the monophasic nature of the pulse will trigger pH modifications and reactions in the tissue. Can you please discuss this issue and quantify it with pH imaging techniques for example?

In POLYRETINA we implemented titanium electrodes. This has an important impact on the stimulation mechanism. This was already discussed a little bit at page 11 lines 17-25 of the former

manuscript. As reported in literature (e.g. doi:10.1146/annurev.bioeng.10.061807.160518) the reactions at the electrode-tissue interface can be capacitive, involving the charging and discharging of the electrode-electrolyte double layer, or faradaic, in which surface-confined species are oxidized and reduced. Materials like Titanium, Titanium Oxide, and Titanium Nitride used in electrodes employ a capacitive charging mechanics, while materials like Platinum and Sputtered Iridium Oxide use a faradic mechanism. Capacitive charge-injection is more desirable than faradaic charge-injection because no chemical species are created or consumed during a stimulation pulse. If only capacitive redistribution of charge occurs, the electrode/electrolyte interface may be modelled as a simple electrical capacitor. Indeed, the capacitive nature of the electrode-electrolyte interface is visible from the decay of the current with prolonged illumination (Fig. 4b, left), while the voltage remains high as expected from a capacitor. In principle, the pixel can be seen as an analogous of a voltage-controlled stimulator.

When the light is turned off, the capacitor is discharged. Therefore, in the absence of any redox (reversible or irreversible) reaction at the electrode-tissue interface (i.e. electron transfer from the electrode material to the electrolyte), the pulse even if monophasic cannot trigger pH modifications in the tissue as supposed for every Faradaic material. We verified this statement with pH measurements, as suggested by the Reviewer. However, we were unable to perform measurements with optical imaging techniques because the green light used to activate the prosthesis cannot be paired in our set-up with a simultaneous fluorescent measure. Therefore, we choose to measure the pH with a pH microelectrode (from UniSense). Results are shown in Supplementary Figure 2 and description is at Page 4 (line 3).

During illumination, a negligible pH shift of about 0.002 pH units has been detected, which could be explained by a recording artefact due to the local temperature increase induced by the prosthesis. Local heating could reduce the resistivity of the solution and decrease the voltage difference between the pH microelectrode and the local reference. Indeed, this pH shift has the same profile of the thermal increase presented in Figure 9.

3. 1000 or 10.000 pulses to look at evolution of the pulse shape is very low (corresponding to 1000s of use for 10Hz stimulation). It is also not clear why 0.5Hz is required as modulation frequency. We can expect very different results from a continuous stimulation at 10Hz compared to alternative stimulation at 0.5Hz envelope. Patients would likely use those devices for a few hours per day which at least means 1 order of magnitude more in continuous use and 2 orders more for chronic use over the years. Can you please provide the evolution of the pulse shape and amplitude over at least 1 day of continuous stimulation (10Hz, 10ms, 1mW/mm² for example - full implant illumination)?

As suggested by the reviewer we now introduced an experiment where the prosthesis is subjected to 320k pulses delivered at 20 Hz (Figure 5e). As can be clearly seen, the current amplitude remains very stable. This is discussed at Page 7 (line 1).

Theoretically, 320k pulses at 20 Hz corresponds to 4.44 hr of operation. In reality, as it is explained in the manuscript, methods section at Page 15 (line 17), our data acquisition system works in blocks of 1 sec. After 1s of stimulation (20 pulses at 20 Hz), it requires 1 sec to save the data and restart the protocol. This mean that our protocols of 320k pulses at 20Hz in reality lasted for 8.88 hours (1 day of operation as suggested by the Reviewer). We wish to clarify here that this is not our intended use of the prosthesis 'in real life', but it is a hardware limitation of the apparatus in use for data acquisition. In addition, the system allows only for a maximum repetition of 1,000 of these 1s (+1 s for data saving) blocks. For this reason, the operator must re-start the sequence every 20,000 stimuli (1,000 blocks of 1s at 20 Hz); This has been repeated 16 times over a working day. That is the reason why we stopped our recordings after 8.88 hr. This time frame is comparable with the battery duration of Argus II, and in general of a device based on camera-glasses. In addition, in Figure 8 we showed that the optoelectronic properties of the prosthesis are not altered after 2 years under accelerated ageing. We are aware that those experiments do not represent perfectly the real life (which could be

represented only by a real use in a patient during a clinical trial); however, we think that together they give enough evidence to support the statement that the prosthesis could operate over a long period of functioning.

4. Figure 3c: why not plotting the PC and PV along with the irradiance instead of using colour-bars? It would provide a visual on the linearity of the charge injection capabilities.

The graphs have been modified accordingly. However, for clarity, since the photocurrent reaches its maximum at 10 ms, there is not much difference in the plot among the peaks with 10/50/100/200 ms of illumination. Therefore, in Figure 4c only the results for 10 ms have been included, while the others (50,100,200) are visible in Figure 4d where the old picture has been moved.

5. Ex-vivo study: The number of retinas and cell used is extremely limited. Only 8 cells for the most interesting experiment (the text mentions only 4 cells but the figure states 8 by the way!!) (figure 5). This is NOT sufficient. It also explains why there are discrepancies between figure 4 and 5. The maximum value for the spiking rate for low latency spikes at 1mW/mm² on figure 5 is the same as the one for 10000uJ/mm² on figure 4!! This clearly shows that the number of experiments is not sufficient and there are big mismatch in the datasets. Both figure 4 and 5 should reflect the same results and be integrated in a single figure and I suspect they were split because of those discrepancies.

We would like first to clarify the mistake in the cell number. In the former manuscript, the cell number was written in a correct manner (n = 8) in the legend. The mistake was in the text as you correctly pointed out. Said that, we agree that a discrepancy exists. This could be due to several reasons, most likely to the biological variability (also because of the low n of the dataset as you suggested). We therefore runned a new batch of experiments with a larger numerosity (n = 39). Moreover, in this new experiments all irradiance values (only for 10 ms pulses) have been tested in the same batch. New results are visible in Figure 6 and Supplementary Figures 3 and 4.

6. I would suggest removing the data from the high power stimulation which is thermally non sustainable anyways and do more experiments at lower irradiances (at least 100 cells).

As suggested we removed the data with pulse duration of (20, 50, and 100 ms) and kept only the data with 10 ms pulses. We ran a new batch of experiments with a numerosity of n = 39. However, we find the request of recording at least 100 cells unjustifiable for any statistical support. We agree that 8 can be considered as too low, but our new data including 39 cells is sufficiently adequate to show the responsivity at increasing irradiances (the error bars are also very small). In other papers such as those on photovoltaic prostheses (e.g. doi:10.1038/nphoton.2012.104), a reasonable number of cells has been considered to be on order of 20. Moreover, our experimental setting requires the use of single metal electrodes for recording data and not MEA devices (as in doi:10.1038/nphoton.2012.104) where 100s of cells can be sampled easily in few retinas. In our epi-retinal configuration, we cannot use an MEA device, and with metal electrodes the number of cell per retinas is much more limited. Moreover, recording 100s of cells will requires a number of animals which cannot be justified to the ethical committee of the Swiss confederation by any statistical mean.

7. Finally, about thermal safety: The assumptions for calculation of safety are wrong.
- the maximum permissible exposure is calculated for a visual angle of 5deg whereas the device itself is designed to provide high visual field (42deg).

We thank the Reviewer for highlighting this aspect. In the new manuscript, we calculated the chronic maximum permissible exposure (MPE) considering the real visual angle (see section Optical and thermal safety at Page 9) and in the methods at Page 11. Indeed, both the photothermal damage and the photochemical damage are depend by alpha (visual angle). The first one increase with α^2 via the factor C_E while the second one is directly related to α^2 . In both cases, as expected, the MPE (in W) increases with the increase of α . Also the illuminated retinal area increases with α^2 . In

summary when computing the MPE in W/mm^2 , we obtained comparable values. The only difference is that now the MPE is limited by photothermal damage.

- the energy absorbed by the implant does not magically disappear. It is converted in heat with less efficient cooling compared to the RPE/retina due to lack of perfusion.

- In those conditions, $1\text{mW}/\text{mm}^2$ is close or above the thermal safety limits.

Please integrate in your assumption the real beam size powering the chip (42deg), the fact that the implant absorbs some energy and is actually heated.

To answer the second question related to the light absorbed and the consequent generation of heating, we measured the change in the surface temperature upon pulsed illumination. Indeed, the thermal safety standards for active implantable medical devices (ISO 14708-1 / EN 45502-1) requires that the maximum temperature on the surface of the implant should not exceed 2°C above the normal surrounding body temperature of 37°C . Our experiment showed that the surface temperature of the device quickly increases and reaches a stable profile after 10 min of stimulation (Fig. 9). The average (\pm s.d., $N = 4$) thermal increase at steady state was $1.24 \pm 0.29^\circ\text{C}$, which is largely below the standard limit of 2°C . Moreover, this experiment corresponds to the extreme case of projecting a constant full white frame, which is not realistic in daily operation when images will be presented as black and white. Under normal daily conditions, the average light dose is lower and therefore the related increase in temperature will be lower. In addition, the eye vitreous has a thermal conductivity ($0.6 \text{ W}/\text{m K}$) ~ 30 times higher than air ($0.02 \text{ W}/\text{m K}$), therefore heat sinking will be more efficient.

Unfortunately, those measures with a IR camera cannot be performed if the device is immersed in solution.

Overall, this paper describes a nice technology but does not provide enough data to demonstrate its usefulness and efficiency.

Thanks for your useful comments, we are sure that in the current revision we have satisfied your concerns.

Reviewer #2 (Remarks to the Author):

Dear Authors,

Congratulation to your work on a wide field photovoltaic prosthesis to cure receptor degenerations. Your device holds the promise to overcome certain limitations of devices which are already on the market.

We would like to thank the Reviewer for her/his positive words.

1. Before I can give my recommendation to publish your manuscript I would like to ask you if you can exactly describe how it will function under real life conditions. You described how defined light stimuli can activate pulse stimulation via the electrodes in your stimulator. You described stimulus frequencies of 1- 10 Hz. How does this fit to a real life condition where images onto the retina are faster. Does your system uses ambient light or is it necessary to wear an external device for projecting a light beam to the desired spot on the implant? So please describe the system architecture.

When introducing photovoltaic stimulation, it is reasonable to ask whether these types of prostheses can work under ambient light conditions. No, the prosthesis cannot be operated by ambient light. This is because of two major reasons: (1) in general, and for sure in the case of POLYRETINA, ambient light entering the pupil is much too dim for photovoltaic stimulation. Moreover, (2) continuous illumination may not be beneficial and eventually detrimental for the photovoltaic system. The titanium electrodes employed in POLYRETINA work under a capacitive principle. Upon light absorption, electrical charges (electrons) accumulate at the electrode-electrolyte interface (electrode side) inducing an ionic charge redistribution in the electrolyte side. Since it works mainly a capacitor, the current generated decreases with time (as shown in Figure 4), therefore implementing continuous light has no benefit for our system. This is why research groups in this field are considering the use of pulsed light, that in humans will be provided through projectors mounted on video goggles. A camera will acquire the ambient and images will be filtered (by a portable processing unit) and converted into an appropriate light stimulus for the prosthesis. This concept has been already proposed in previous research papers, mainly from the research group of Prof. Palanker at Stanford.

Some other issues are listed here. I wish that you can comment on each of these points.

2. Page 2, first passage: The definition of blindness depends on the country where you are. So please indicate for what country this definition is valid.

We thank the Reviewer for this observation. We indeed modified the sentence into a more precise statement. Here is reported in italic the new sentence at Page 2 (line 1): *'Blindness affects more than 30 million people worldwide, and it is defined as visual acuity of less than 20/400 or a corresponding visual field loss to less than 10 degrees, in the better eye with the best possible correction (WHO, ICD-11). In North America and most of European countries, legal blindness is defined as visual acuity of 20/200 or visual field no greater than 20 degrees.'*

3. These diseases are rare diseases. Degenerative diseases such as macular degeneration, glaucoma or diabetes are much more frequent reasons for blindness. The statement could be misunderstood. It sounds that it is easy to identify patients suitable for the implantation. In reality it is extremely difficult to find appropriate patients for an implantation.

To avoid any misleading information, we changed the sentence at Page 2 (line 4) as follow: *'In the last decade, various visual prostheses have been developed to fight blindness in case of retinal dystrophies, such as Retinitis pigmentosa2 and more recently age-related macular degeneration (Clinical Trial NCT02227498).'*

4. Page 2, line 25: The stiffness depends on the thickness. Using very thin polyimid e.g. could result in a very flexible substrate.

This is indeed right. We changed the sentence at Page 2 (line 21) in order to be scientifically more accurate. Here is also reported: *‘However, these approaches are based on materials (i.e. polyimide) with high elastic modulus (~GPa), very thin substrates (e.g. 10 μm), and complex shapes (e.g. star) that could create challenges in manipulation, implantation, and fixation’.*

5. Page 3, line 23: The inner surface of the eye does not have a simple spherical shape. The retinal surface could vary individually with many different local radii.

We thank the Reviewer for this very important observation. However, here we only stated that the process (molding) is so simple that the prosthesis can be fabricated according to any realistic design. We think that our message is scientifically accurate, since (provided the real curvature/shape of the patient eye) we can fabricate prostheses with any radius of curvature and/or with non-uniform spherical shape. The sentence, now at Page 13 (line 4) has been modified as follow: *‘However, the flexibility in the fabrication process of the dome-shaped PDMS support (PDMS molding) allows the fabrication of prostheses designed to fit the real eye curvature/shape of a patient’.*

6. Page 4, line 8: PDMS may also have a disadvantage in terms of its failure to seal electronics from water on the long run.

Yes, PDMS has some disadvantages. However, there are few aspects that it is important to clarify. This device is a passive photovoltaic device, in the sense that there is no power supply or active electronics embedded. Therefore, the problem of sealing electronics from water is not an issue, since it has been already demonstrated that organic passive devices can work in liquid environment without sealing. PDMS is very permeable to gases and much less to water. Moreover, PDMS is already in use for ‘sealing’ clinical electrode arrays, such as the one made by CorTec (<http://cortec-neuro.com>) or cochlear implants. The main advantage of PDMS is that it is available as Medical Grade.

7. Page 5, line 1: How close is this model to the real situation? Can you provide images showing the close approximation of the device with the retina? Page 5, line 2-4: Do you already have any experiences with the implantation in animals? The implantation of large devices through a large opening may result in introducing the device during a hypotonous phase which makes the implantation procedure relatively uncontrolled. Plastic models are not suitable to simulate this situation because these models do not collapse.

The experiments with plastic eyes were very useful to provide a first ‘draft’ of surgical approach and validate the capability of the prosthesis to fold and unfold, which is one of the main statement of the paper. The implantation in living animal is planned for Q1 of 2018. Dr. Prof. Thomas J. Wolfensberger (author of this paper) will lead this phase. Currently we are focusing on the surgical optimization in enucleated pig eyes. Since it has been specifically requested by the Reviewer we added a figure showing the same approach performed in freshly enucleated pig eyes (Fig. 2). This should clarify the doubt about the use of plastic eyes. However, we would like to clarify that we added the figure as illustrative example, since the optimization of the surgical approach is still ongoing.

8. Page 7, line 5: With such stimulus frequencies moving objects can not be sufficiently perceived. Usually image frequencies of 24-25 Hz are used to obtain perception of moving objects. Simultaneously the stimulation of ganglion cells may require much higher stimulus frequencies with very short pulses. The system will loose its flexibility if it is limited to a low stimulation rate.

In the revised manuscript we show that the prosthesis can safely work at 20 Hz; and all our estimation (illumination safety, temperature, pH etc) are now based on this frequency. As we clarified in the discussion at Page 11 (line 15), 20 Hz is not far from the operation frequency of other retinal prostheses such as Argus II and Retinal Implant AG.

9. Page 10, line 18: Again, PDMS is not a safe seal.

At this line we mainly refer to a real mechanical effect. We observed in the past that if the polymer cracks it may delaminate easily if exposed to liquid, while the layer of PDMS protects from mechanical delamination.

10. Page 11, line 8-10: It is true that according to ISO 10993 the mentioned tests have to be performed. However, it would be great to see that the device can be safely placed in eyes of mid-size animals and only then (clinical exams, histology, evidence for cortical activation) one can decide if the large device is suitable for implantation or not.

As we wrote before this is a planned study that will start in January 2018. However, the Reviewer is aware that providing clinical exams, histology, and evidence for cortical activation is a project that will require 2 to 3 years to be completed; this will require a dedicated animal experimentation license, the validation of the surgical approach, the validation of a blind model of pig (since we use visible light), and the experimentation. We think this is beyond the scope of this manuscript.

Reviewer #3 (Remarks to the Author):

Ferlauto, Airaghi Leccardi, Chenais et al. present a new design for a foldable photovoltaic epiretinal prosthesis. Their device is flexible and can consequently be implanted through a relatively small scleral incision. The implant is curved so as to remain in tight contact with the retina over a large area, theoretically enabling electrical activation of the retina over a wide field of view in blind patients (42 degrees, significantly larger than existing implants). The authors detail manufacturing techniques and characterize their implant in-vitro using test devices and ex-vivo using a mouse model of retinal degeneration.

While the manuscript is well written and has the potential to appeal to the wider medical implants community where flexible electronics are becoming increasingly relevant, in addition to the retinal prosthesis community, some claims made by the authors need better substantiation, especially the claims regarding high visual acuity and high fidelity in retinal stimulation.

1. Regarding high visual acuity: at the abstract level and in the discussion (p.11 l.14-16), the authors state that their device represents a solution to improve the visual acuity of retinal prostheses, by “increasing the density of stimulation sites”. However, in the densest areas of the implant (5-mm diameter central area), the device only embeds 967 pixels, which should translate to an electrode pitch of roughly 130 μ m, assuming uniform coverage of the central area (it would be helpful to know the actual manufactured electrode pitch).

The electrode spacing has been added in Supplementary Figure 1 and in the legend of Figure 1, where the geometry of the array is described. The pitch in the central area is 150 μ m.

2. The electrode density of this device must then be four times lower than the current densest devices already described in the literature (Alpha IMS subretinal implant by Retina Implant AG, which should be mentioned in the introduction, pixel pitch on the order of 60 μ m, or Stanford photovoltaic implant, cited in refs 12 and 13, pixel pitch on the order of 65 μ m). From Nyquist sampling considerations, the best visual acuity this device can hope to achieve is on the order of 20/500, still below the legal limit for blindness (20/200 USA, 20/400 WHO), and a figure that's already been achieved by some of the patients of the Alpha IMS patient cohort. The language in the text should therefore be moderated accordingly.

Here the Reviewer is comparing an epi-retinal prosthesis with respect of two devices for subretinal implantation (Alpha IMS and Stanford/Pixium silicon photovoltaic implant). We do not think that the number of pixel and density plays the same role in epiretinal and subretinal stimulation. A fair comparison will be with other devices for epiretinal implantation where the number of pixels is in the order of tens (Second Sight, 60 electrodes) or hundreds (diamond epiretinal prosthesis in Melbourne, 256 electrodes). In this scenario, our statement (p.11 l.14-16; former manuscript) has to be considered as improvement with respect to the ‘comparable’ devices. According to the reviewer request, we changed the sentence (Page 10 Line 15) as follow: *‘One of the most important open questions in the field of retinal prostheses concerns how to increase both visual acuity and visual field size together. From the engineering point of view this implies to increase the density of the stimulating electrodes and enlarge the size of the prosthesis. POLYRETINA is a novel foldable and photovoltaic wide-field epiretinal prosthesis with a remarkable increase in its size (46.3 degrees) and in the number of stimulating pixels (2215) compared to other epiretinal prostheses’*. We also moderate the text accordingly through the entire manuscript.

3. Additionally, recent evidence seems to indicate that the number of pixels required to perform tasks such as recognition of daily life objects is closer to a few thousands than the few hundreds cited in the text (figure from p.2, l.28, refs 10, 11): see the work of Jung et al., Vision Res. 2015 Jun; 111(0 0): 182–196. It therefore seems that the electrode density needs to be significantly increased before the device can claim to hope to restore vision with high visual acuity. That being said, the

photovoltaic design shown here appears to be highly scalable, so it could be sufficient to discuss what it would take to achieve a higher electrode density.

We agree with the Reviewer. First this important paper has been now cited appropriately in the text (Page 2, Line 25). Second, the Reviewer is right when (s)he said that the technology is very scalable. Indeed, in the lab we are working on denser design. Currently we are performing mechanical simulation and characterizations; this experiments will be followed by functional characterizations next year. This is preliminary activity which we think is beyond the scope of the present manuscript; they will be the core results of a new manuscript.

As suggested by the Reviewer we integrated in the discussion a paragraph explaining how higher electrode density can be achieved. This is at Page 11 (Line 5). Here it is reported for the Reviewer: *'Concerning visual acuity, with a pitch of 150 μm the theoretical visual acuity restored by POLYRETINA is in the order of 20/600; which is better than the current epiretinal prostheses (e.g. Argus II) but still below the threshold of legal blindness. However, the technology of POLYRETINA is highly scalable. Based on mechanical simulations (not shown), the pitch can be reduced down to a value of 110 μm , keeping the same electrode size (80 μm), thus approaching the theoretical value of 20/400. A further improvement consists in reducing the size of the electrode (i.e. 60 μm) with a pitch of ~80 μm , thus approaching a theoretical visual acuity of 20/300, similar to the silicon photovoltaic subretinal prosthesis¹⁴. However, these values come from theoretical computation, and therefore must be validated with proper in-vivo experiments in animals and later in humans. Moreover, the reduction of the pixel size will reduce the PC generated by the interface, therefore the efficiency in stimulating RGCs should be validated again'.*

4. Increased electrode density does not however necessarily translate into improved visual acuity, so more confidence that the device behavior is currently limited by its pixel pitch and not flaws in design is required. For example, cross-talk between neighboring electrodes because of inadequate grounding, or because of diffusion of photovoltaically-induced carriers between one pixel and the next could all contribute to a degradation of the implant performance in practical situations. From the design shown in Supplementary Figure S1, it is unclear how one could achieve focal stimulation with the current device, for the two following reasons.

(1) Assuming that tight contact is maintained between the retina and the implant, currents would then get channeled laterally around the device back to the PEDOT anode, leading to strong cross-talk between neighboring pixels. (2) With a single layer of P3HT:PCBM, what's to prevent photocarriers from diffusing away from the target pixel and actually be injected at the site of the neighboring electrode? Silicon photovoltaic implants (refs 12, 13) include physical gaps or silicon oxide trenches between pixels to prevent diffusion of the photocarriers. Some modeling work, or current mapping similar to what was done by Flores et al., Journal of Neural Engineering 13: 036010 would give confidence that the device could theoretically be able to sustain high-resolution percepts. The modeling work would ideally clarify to what extent cross-talk should be expected between adjacent pixels (from carrier diffusion and the grounding scheme), and whether small pixels (ideally at least as small or smaller than the Alpha IMS and Stanford photovoltaic implant) can provide enough charge to stimulate the retina despite a decreased light-sensitive area, given a realistic estimated distance between the implant and the retina. Published current values in the literature can be inadequate, especially when electrode sizes vary, and distance between cells and electrodes are not the same as those of the original study.

The Reviewer is concerned, given the continuous BHJ blend in the device, that photogenerated charges can travel laterally and cause interference in adjacent Ti electrodes.

This would certainly be an important factor in traditional inorganic photovoltaics (e.g. silicon) where free carriers are generated directly upon light absorption and diffuse long distances (even up to mm) in the film. This is why, in silicon devices (e.g. Silicon Retina developed in Stanford by Prof. Palanker) physical gaps or silicon oxide trenches are necessary between pixels. However, in the organic semiconductor P3HT:PCBM blend used in this work, it is well known that the active area of

the device is defined by the area of the cathode (usually Al, but Ti is used in this case) due to the poor (hopping based) transport of free charge carriers. Indeed, in organic photovoltaics, the low carrier mobility and lifetime limit the carrier-transport length to tens of nm for holes and few hundreds of nm for electrons (doi:10.1103/PhysRevB.82.205325). Experimentally it has been shown by another group that the photocurrent detected at the cathode is reduced to ~10% of the maximum if the illumination spot (size 1 μm) is moved laterally by ~12 μm from the electrode edge (doi:10.1063/1.2998540). This large decay length, beyond the simple diffusion processes, has been explained by a steady state nonlocal electric-field inducing a lateral flow of the separated carriers. For this reason, an internal cross-talk between electrodes due to charge carriers generated under one electrode traveling laterally towards an adjacent electrode can be excluded (at least down to a edge-to-edge electrode distance of ~20 μm). This is why POLYRETINA does not require trenches between pixels. This explanation has been introduced in the new manuscript at Page 10 (line 23). Following the suggestion of both Reviewer 1 and 3, we also performed an experiment coupled by FEA simulations. These additional evidences shows that it is possible to claim a spatially selective stimulation with the current device. These new evidences are included in Figure 7 and described at Page 8 (line 6).

As reported in the previous reply (Point 3), in the lab we are working on denser design. Currently we are performing mechanical simulation and characterizations; this experiments will be followed by functional simulation and characterizations next year. This is preliminary activity and we think that the electrical characterization with smaller and denser pixels is beyond the scope of the present manuscript; they will be the core results of a new manuscript.

5. Regarding fidelity of retinal stimulation: in terms of physiological testing, some choices of stimulation parameters in the ex-vivo evaluation are surprising. It seems unnecessary to show retinal responses for stimulation irradiances above around 10 mW/mm² when safety considerations for chronic exposure will limit the available power to 7.77 mW/mm² or lower, depending on the pulse rate (e.g. Supplementary Figures 2 and 3). The measurements taken between 10 and 100 mW/mm² were taken with repeated stimuli, so they depart from the assumptions made in the single pulse exposure calculation to show safety on p.9 (it's good to know though that single pulse limits are so high, so I am not suggesting removing this derivation).

In the new manuscript we included a new batch of cells obtained only with 10 ms, for a large scale of irradiance. This allow us to provide a better plot of the spike frequency vs irradiance, as suggested by Reviewer 1. New values of safety limits are also reported. We think that now the data are much more in line with the Reviewer expectation.

6. In places, thresholds are reported in terms of energy deposited onto the implant (e.g. p.8, but also Supplementary Figure 6). It is useful to know both what pulse width and what irradiance were used, and not just energy, as stimulation thresholds depend jointly these two parameters through the strength-duration relationship for the stimulated tissue.

In the new manuscript we only included data obtained with 10 ms pulses, therefore this issue is not present anymore. The choice of 10 ms pulses is coherent with the demonstration that the prosthesis can operate at 20 Hz, which is 1 pulse every 20 ms.

7. In the discussion, the authors mention (p.12, l.20-23) the high fidelity in retinal stimulation of their implant. However, reproducing the retinal code with high fidelity requires stimulation at frequencies of up to several hundred Hz (see for example Fried et al., J Neurophysiol 95: 970–978, 2006), which does not seem attainable with a photovoltaic implant that passively discharges between pulses. Boinagrov et al., IEEE Trans. Biomed. Circuits and Systems 10(1): 85-97, 2016 suggests that a shunt resistor is required to help discharge stimulation electrodes between pulses for a photovoltaic devices activated at a few 10s of Hz, which is missing in this implant.

In the new manuscript we show that our implant is capable to work at 20 Hz of frame rate, also without a shunting resistance. This is because of the different shunting properties of a thin film of polymer with respect to silicon. In the discussion at Page 11 (Line 15), we reported that 20 Hz is within the operative range of several other retinal prosthesis, including argus II.

Regarding 'fidelity' our statement was related to the small jitter we found in the activation of SL spikes. We were not referring to the reproduction of the natural retinal code. The statement has been removed in the new manuscript.

8. Additionally, l.23-25, the authors mention that ML and LL spikes contain signals coming from activation of passing axons. This is incorrect: axonal activation also has a short latency, see Grosberg et al., J Neurophys DOI: 10.1152/jn.00750.2016 for a detailed description of electrical signatures of axonal activation. Independently of this issue, the evidence in the literature currently points to benefits of long pulses being delivered epiretinally to target the inner retina and bypass the RGC layer. For example, Weitz et al. in Sci Transl Med. 2015 Dec 16;7(318):318ra203 show they can improve the resolution of an epiretinal array by increasing stimulus duration. This study should be cited in this section of the discussion section (p.12 l.20 and after). It appears to me that the authors would want to optimize activation of the INL without activation of the RGC and axon layer instead of trying to selectively activate the RGCs only, unless they can come up with another method for bypassing axonal activation (which seems hard with short pulses, see Grosberg et al. 2017).

We agree that the previous discussion on this topic was misleading. First we now add in the discussion at Page 12 (Line 10) the paper from Weitz and co-authors. We also compare our results with their results, and we explain how the activation of ML and LL spikes due to the indirect activation of RGC could be beneficial for more focal activation. The text is reported here in italic: *'On the contrary, ML and LL spikes are due to the activation of the internal retinal circuit. It is known that brief (hundreds of μ s) cathodic epiretinal stimulation preferentially excite RGCs, while longer pulses (> 1 ms) excite both RGCs and bipolar cells. It has been recently demonstrated that the use of pulses shorter than 8 ms results in the activation of axons of passage that causes streak responses, while longer pulses results in a more focal activation. Using calcium imaging techniques, authors explained this result via a shift from direct to indirect activation of RGCs. We showed by electrophysiological recordings and pharmacological experiments that the cathodic stimulation provided by POLYRETINA is also indirectly activating RGCs. This represents a promising result for the in-vivo translation of POLYRETINA in order to obtain a focal activation. Further experiments aiming at dissecting the circuit activated by POLYRETINA will help in defining the appropriate stimulation parameters to obtain a more focal stimulation'*.

Reviewers' comments:

Reviewer #1 (Remarks to the Author):

The authors gave precise and clear answer to most of the questions raised by the reviewers, the paper has been greatly improved. However, there are still points requiring clarification:

Q3. I understand that the recording system to look at the evolution of the pulse shape and amplitude with time has its limitation, it's perfectly fine. However, the stimulation system should not have. The light source could easily be driven independently by a pulse generator continuously and you could record one of the pulses every second or so. This may or may not make a big difference but it would provide much more convincing data.

Q5. The number of cells is large enough now and the data looks more convincing.

Q7. Safety claims are critical as once they are published, they become somehow gold standard with no further verification. Here I am sorry to say that the claim of optical safety is not substantiated and should be removed if not more convincing evidence is provided. Indeed the authors show a temperature increase of the device alone of 1.24 ± 0.29 degC (which is NOT "far bellow" 2 degC) when only half of the light is absorbed by the implant. The rest of the light is absorbed by the choroid and RPE and induces further heating and those effects accumulate. Here the authors assume independence of the 2 effects and rely on the simplistic ANSI standard which will not apply here.

If the authors decide to remove all claim about optical safety it is fine, but if they want to maintain them, they will need to estimate quantitatively the cumulative effect of heating the implant and the retina.

Reviewer #2 (Remarks to the Author):

Thank you for responding to my concerns and remarks.

I thin you responded well to each of these issues and therefore I will recommend publication.

Sincerely

Reviewer #3 (Remarks to the Author):

In this revision, Ferlauto, Airaghi Leccardi, Chenais addressed most of my concerns, and significantly improved the strengths of the claims made in their manuscript. However, some additional clarity is still needed on the following two points (I do not expect these clarifications to change the substance or value of the manuscript):

- (1) Optical safety limits; and
- (2) Origins of the elicited spikes vs. latency.

(1) Safety limits.

Regarding safety limits (text on page 9), some of the exposition by the author is confusing, and I am not sure it is accurate either. Specifically, the following statements need attention:

- "Regarding POLYRETINA, the maximum permissible exposure (MPE) during chronic illumination of the full prosthesis (equivalent to a full white frame) is controlled by the photothermal damage and equal to $328.75 \mu\text{W}/\text{mm}^2$ (see Methods)" [...]
- "Therefore, the overall MPE could be increased to $669.97 \mu\text{W}/\text{mm}^2$ "

Energy absorbed by the POLYRETINA does not disappear after it is absorbed, and thus will contribute to heating of the eye (as the authors rightly point out on line 27). Why would the MPE then be increased to $670 \mu\text{W}/\text{mm}^2$ in the presence of the device? This is not a situation where an ND filter prevented half of the incident energy from making its way to the back of the eye, and two additional factors governed by the (nonlinear) heat diffusion equations need to be taken into account rigorously.

The first effect is that for half of the incident energy, the location where it's been absorbed has merely been shifted by $\sim 150 \mu\text{m}$ towards the lens (from the RPE/choroid to the POLYRETINA). This is likely to keep the same or slightly decrease, and not increase the MPE. Choroidal blood flow can actively cool down the retina when its temperature increases, thanks to increased convective cooling caused by increased blood perfusion. By displacing the location where heat is absorbed away from this active cooling mechanism, it is likely heat will be moved away from the retina less efficiently, which should result in a decreased MPE. Heat always diffuses into the vitreous from the retina, and this is taken into account by existing standards, so even if the vitreous is more heat conductive than air it will not help.

The second effect is that the thermal conductivity of the environment right above the retina goes from being that of the vitreous, to that of the POLYRETINA. If the polyretina is a very effective heat conductor, this can help increase the MPE by efficiently diffusing heat away from the illuminated zone. If it is a worse heat conductor than the vitreous, instead, the MPE will likely be lower than without the implant.

Either way, these effects cannot simply be taken into account by applying a 50% scaling factor to the MPE in the absence of the polyretina. I believe the modeling works done in ref. 31 of the paper should apply directly to the question of thermal safety, and similar FEM simulations of thermal safety should be performed to adequately put this question to rest.

(2) Elicited spikes vs. latency and discussion of visual acuity

The following point is more minor than the question of thermal safety. In the results section (page 8), the authors explain that they "verified in a second subset of cells ($n = 6$, $N = 5$; 209.4 ± 37.14 days) that the prosthetic activation of both ML and LL spikes is abolished by using synaptic blockers".

The sample size here is rather modest, but given the results already present in the literature (including some papers already cited in the article), it would be extremely surprising to witness direct activation with latencies longer than 10 ms. Rather, it is the classification of the SL spikes that leaves to be desired: not all spikes with latencies < 10 ms originate directly in the ganglion cells (see ref. 27 for example). This would be worth clarifying: there remains uncertainty in the results concerning the origin of the SL spikes reported by the authors.

We would like to thank all the Reviewers for the careful revision and the interesting comments. We found them very useful to improve our manuscript. Answers have been highlighted in blue.

Reviewer #1: The authors gave precise and clear answer to most of the questions raised by the reviewers, the paper has been greatly improved. However, there are still points requiring clarification:

Q3. I understand that the recording system to look at the evolution of the pulse shape and amplitude with time has its limitation, it's perfectly fine. However, the stimulation system should not have. The light source could easily be driven independently by a pulse generator continuously and you could record one of the pulses every second or so. This may or may not make a big difference but it would provide much more convincing data.

This is correct, we could have done it in this way. However we do not think that it substantially change the statements in the manuscript or provide much more convincing data. In future, we would like to perform these recordings under accelerated ageing to provide a more complete estimation of the lifetime. However, this may require several months to be completed.

Q5. The number of cells is large enough now and the data looks more convincing.
We are happy that the Reviewer found our data convincing.

Q7. Safety claims are critical as once they are published, they become somehow gold standard with no further verification. Here I am sorry to say that the claim of optical safety is not substantiated and should be removed if not more convincing evidence is provided. Indeed the authors show a temperature increase of the device alone of 1.24 ± 0.29 degC (which is NOT "far below" 2 degC) when only half of the light is absorbed by the implant. The rest of the light is absorbed by the choroid and RPE and induces further heating and those effects accumulate. Here the authors assume independance of the 2 effects and rely on the simplistic ANSI standard which will not apply here. If the authors decide to remove all claim about optical safety it is fine, but if they want to maintain them, they will need to estimate quantitatively the cumulative effect of heating the implant and the retina.

Following the suggestion of the Reviewer (and also Reviewer 3), we included FEA simulations showing the temperature variation in the retina upon illumination of the prosthesis. In this case, we performed simulations similar to what has been proposed in ref. 31. The results are visible at page 10 of the updated manuscript, in Figure 10, and in Supp. Figures 7 and 8. To summarize, we observed that the presence of the implant slightly decrease (~11%) the overall retinal heating. This means that the MPE indeed increases (with respect to the case without POLYRETINA) but only by a very small factor. We also removed the "far below" from the manuscript. We think now the statement is corrected and clarified.

Reviewer #2: Thank you for responding to my concerns and remarks. I thin you responded well to each of these issues and therefore I will recommend publication. Sincerely
We would like to thank the Reviewer for her/his positive feedback.

Reviewer #3: In this revision, Ferlauto, Airaghi Leccardi, Chenais addressed most of my concerns, and significantly improved the strengths of the claims made in their manuscript. However, some additional clarity is still needed on the following two points (I do not expect these clarifications to change the substance or value of the manuscript):

- (1) Optical safety limits; and
- (2) Origins of the elicited spikes vs. latency.

- (1) Safety limits.

Regarding safety limits (text on page 9), some of the exposition by the author is confusing, and I am not sure it is accurate either. Specifically, the following statements need attention:

- “Regarding POLYRETINA, the maximum permissible exposure (MPE) during chronic illumination of the full prosthesis (equivalent to a full white frame) is controlled by the photothermal damage and equal to $328.75 \mu\text{W}/\text{mm}^2$ (see Methods)” [...]
- “Therefore, the overall MPE could be increased to $669.97 \mu\text{W}/\text{mm}^2$ ”

Energy absorbed by the POLYRETINA does not disappear after it is absorbed, and thus will contribute to heating of the eye (as the authors rightly point out on line 27). Why would the MPE then be increased to $670 \mu\text{W}/\text{mm}^2$ in the presence of the device? This is not a situation where an ND filter prevented half of the incident energy from making its way to the back of the eye, and two additional factors governed by the (nonlinear) heat diffusion equations need to be taken into account rigorously.

The first effect is that for half of the incident energy, the location where it's been absorbed has merely been shifted by $\sim 150 \mu\text{m}$ towards the lens (from the RPE/choroid to the POLYRETINA). This is likely to keep the same or slightly decrease, and not increase the MPE. Choroidal blood flow can actively cool down the retina when its temperature increases, thanks to increased convective cooling caused by increased blood perfusion. By displacing the location where heat is absorbed away from this active cooling mechanism, it is likely heat will be moved away from the retina less efficiently, which should result in a decreased MPE. Heat always diffuses into the vitreous from the retina, and this is taken into account by existing standards, so even if the vitreous is more heat conductive than air it will not help.

The second effect is that the thermal conductivity of the environment right above the retina goes from being that of the vitreous, to that of the POLYRETINA. If the polyretina is a very effective heat conductor, this can help increase the MPE by efficiently diffusing heat away from the illuminated zone. If it is a worse heat conductor than the vitreous, instead, the MPE will likely be lower than without the implant.

Either way, these effects cannot simply be taken into account by applying a 50% scaling factor to the MPE in the absence of the polyretina. I believe the modeling works done in ref. 31 of the paper should apply directly to the question of thermal safety, and similar FEM simulations of thermal safety should be performed to adequately put this question to rest.

Following the suggestion of the Reviewer (and also Reviewer 1), we included FEA simulations showing the temperature variation in the retina upon illumination of the prosthesis. In this case, we performed simulations similar to what has been proposed in ref. 31. The results are visible at page 10 of the updated manuscript, in Figure 10, and in Supp. Figures 7 and 8. To summarize, we observed that the presence of the implant slightly decrease ($\sim 11\%$) the overall retinal heating. This means that the MPE indeed increases (with respect to the case without POLYRETINA) but only by a very small factor.

(2) Elicited spikes vs. latency and discussion of visual acuity

The following point is more minor than the question of thermal safety. In the results section (page 8), the authors explain that they “verified in a second subset of cells ($n = 6$, $N = 5$; 209.4 ± 37.14 days) that the prosthetic activation of both ML and LL spikes is abolished by using synaptic blockers”.

The sample size here is rather modest, but given the results already present in the literature (including some papers already cited in the article), it would be extremely surprising to witness direct activation with latencies longer than 10 ms. Rather, it is the classification of the SL spikes that leaves to be desired: not all spikes with latencies < 10 ms originate directly in the ganglion cells (see ref. 27 for example). This would be worth clarifying: there remains uncertainty in the results concerning the origin of the SL spikes reported by the authors.

We agreed that the sampling is limited, however as the Reviewer also pointed out, this is an ‘obvious’ control since the effect of synaptic blockers has been already reported in many papers. About the latency of directly activated spikes, SL spikes are reported to be very close to the stimulus (i.e. 0.5-4

ms). In Ref 27, it is claimed that SL responses do not exceed 5 ms. However, all the previous papers evaluated those latencies with electrical stimuli which are typically sharp squared pulses (rising slope ideally infinite). In our case the photo voltage/current generated by the prosthesis has a less shaper transition from 0 to the peak. We showed that the peak is reached in about 10 ms. Therefore also latencies may be affected and we cannot assume that the values obtained by sharp current pulses can be used as reference. Despite this uncertainty, however in our data we mostly observe only 1 spike in the first 10 ms. Rarely multiple spikes are present. The average latency is about 4.12 ms, therefore we are confident that we are actually detecting directly evoked spikes and ML spikes are not present in the 10 ms bin.

Reviewers' comments:

Reviewer #1 (Remarks to the Author):

The authors addressed all the questions. I would recommend the paper for publication.

Reviewer #3 (Remarks to the Author):

This revision addressed my remaining concerns about optical safety of the device and origin of the elicited spikes, and the paper now appears to be adequate for publication.

I would suggest adding the explanation the authors gave for the long latency of direct RGC spikes in their latest rebuttal to the discussion section. The inattentive reader might otherwise fail to notice the slow photovoltage/current rise, and would wonder why spikes with latency 4-10 ms are classified as direct responses.

We would like to thank all the Reviewers for the careful revision and the interesting comments. We found them very useful to improve our manuscript. Answers have been highlighted in blue.

Reviewers' comments:

Reviewer #1 (Remarks to the Author): The authors addressed all the questions. I would recommend the paper for publication.

We thank the Reviewer.

Reviewer #3 (Remarks to the Author): This revision addressed my remaining concerns about optical safety of the device and origin of the elicited spikes, and the paper now appears to be adequate for publication. I would suggest adding the explanation the authors gave for the long latency of direct RGC spikes in their latest rebuttal to the discussion section. The inattentive reader might otherwise fail to notice the slow photovoltage/current rise, and would wonder why spikes with latency 4-10 ms are classified as direct responses.

As suggested by the Reviewer, this explanation has been included in the discussion.

REVIEWERS' COMMENTS:

Reviewer #3 (Remarks to the Author):

The paper now seems ready for publication.